# Kidney organoids recapitulate human basement membrane assembly in health and disease

**Mychel RPT Morais**[1†], **Pinyuan Tian**[1†], **Craig Lawless**[1], **Syed Murtuza-Baker**[2], **Louise Hopkinson**[1], **Steven Woods**[3], **Aleksandr Mironov**[4], **David A Long**[5], **Daniel P Gale**[6], **Telma MT Zorn**[7], **Susan J Kimber**[3], **Roy Zent**[8], **Rachel Lennon**[1,9]*

[1]Wellcome Trust Centre for Cell-Matrix Research, University of Manchester, Manchester, United Kingdom; [2]Division of Informatics, Imaging and Data Sciences, University of Manchester, Manchester, United Kingdom; [3]Division of Cell Matrix Biology and Regenerative Medicine, University of Manchester, Manchester, United Kingdom; [4]Electron Microscopy Core Facility, University of Manchester, Manchester, United Kingdom; [5]Developmental Biology and Cancer Programme, University College London, London, United Kingdom; [6]Department of Renal Medicine, University College London, London, United Kingdom; [7]Department of Cell and Developmental Biology, University of São Paulo, São Paulo, Brazil; [8]Department of Medicine, Vanderbilt University Medical Center, Nashville, United States; [9]Department of Paediatric Nephrology, Royal Manchester Children's Hospital, Manchester University Hospitals NHS Foundation Trust, Manchester, United Kingdom

**\*For correspondence:**
Rachel.Lennon@manchester.ac.uk

†These authors contributed equally to this work

**Competing interest:** The authors declare that no competing interests exist.

**Abstract** Basement membranes (BMs) are complex macromolecular networks underlying all continuous layers of cells. Essential components include collagen IV and laminins, which are affected by human genetic variants leading to a range of debilitating conditions including kidney, muscle, and cerebrovascular phenotypes. We investigated the dynamics of BM assembly in human pluripotent stem cell-derived kidney organoids. We resolved their global BM composition and discovered a conserved temporal sequence in BM assembly that paralleled mammalian fetal kidneys. We identified the emergence of key BM isoforms, which were altered by a pathogenic variant in *COL4A5*. Integrating organoid, fetal, and adult kidney proteomes, we found dynamic regulation of BM composition through development to adulthood, and with single-cell transcriptomic analysis we mapped the cellular origins of BM components. Overall, we define the complex and dynamic nature of kidney organoid BM assembly and provide a platform for understanding its wider relevance in human development and disease.

## Editor's evaluation

Kidney organoid cultures derived from human induced pluripotent stem cells represent a new tool with which to study renal morphogenesis in both normal and pathological states. In the current study, the authors have combined morphological evaluation with proteomics to elucidate aspects of the temporal sequence of basement membrane composition during normal renal development and in the setting of a pathogenic collagen type IV α5 chain variant associated with Alport syndrome, an inherited kidney disease. This model system may help us to better understand the normal processes of basement membrane development and the pathogenesis of inherited diseases that affect renal basement membrane composition.

## Introduction

Basement membranes (BMs) surround tissues providing cells with an interface for physical and signaling interactions (*Jayadev and Sherwood, 2017*). They are composed of laminins, collagen IV, nidogens, heparan-sulfate proteoglycans (*Kruegel and Miosge, 2010*), and many minor components that combine to form biochemically distinct BMs across different tissues (*Randles et al., 2017*). BMs play active morphogenic roles that are critical for tissue and cell fate specification (*Kyprianou et al., 2020*; *Li et al., 2003*), and variants in BM genes are associated with a broad range of human diseases (*Chew and Lennon, 2018*; *Gatseva et al., 2019*). Despite the increasing knowledge of BM composition and function, there is limited understanding about BM regulation, yet this is required for new mechanistic insights into BM-associated human disease.

BMs form early in embryogenesis through binding interactions with cell surface receptors (*Miner and Yurchenco, 2004*) and typically an initial laminin network is required for further incorporation of collagen IV, nidogen, and perlecan into nascent BMs (*Jayadev et al., 2019*; *Matsubayashi et al., 2017*), thus following an assembly hierarchy for *de novo* BM formation. BMs are also highly dynamic, remodeling during morphogenesis to form tissue-specific BMs (*Bonnans et al., 2014*), such as the glomerular basement membrane (GBM) in the kidney, which functions as a size-selective filter. Situated between podocytes and endothelial cells in the glomerular capillary wall, the GBM is formed by the fusion of separate podocyte and endothelial BMs, and further remodeled into a mature GBM. This involves replacement of laminin α1β1γ1 (termed laminin-111) and collagen IV α1α1α2 networks by laminin-511 then -521, and collagen IV α3α4α5 (*Abrahamson et al., 2013*; *Abrahamson and St John, 1993*). These transitions are important for long-term GBM function, and genetic variants in *COL4A3, COL4A4,* and *COL4A5* or the laminin gene *LAMB2* cause defective GBMs and human diseases (*Barker et al., 1990*; *Zenker et al., 2004*).

The study of BM assembly is challenging due to the technical difficulties in tracking large, spatio-temporally regulated components. Most understanding about vertebrate BMs comes from immunolocalization and genetic knockout studies (*Abrahamson et al., 2013*), and for composition, mass spectrometry (MS)-based proteomics has enabled global analyses (*Naba et al., 2016*; *Randles et al., 2015*). Proteomics also allows time course studies, which have provided insight into matrix dynamics during development and in disease progression (*Hebert et al., 2020*; *Lipp et al., 2021*; *Naba et al., 2017*). However, proteomics lacks the spatial context that is captured by localization studies, including fluorescent tagging of endogenous proteins. Such investigations in *Drosophila melanogaster* and *Caenorhabditis elegans* have unraveled dynamic features of BM assembly in embryogenesis and repair (*Howard et al., 2019*; *Keeley et al., 2020*; *Matsubayashi et al., 2020*). The development of a system to study human BM assembly would thereby facilitate investigation of both morphogenesis and disease.

Kidney organoids generated from pluripotent stem cells (PSCs) contain self-organized 3D structures with multiple kidney cell types and represent an attractive system for investigating early development (*Combes et al., 2019b*; *Takasato et al., 2015*). Organoids derived from induced PSCs (iPSCs), reprogrammed from patient somatic cells, have further use in personalized disease modeling and therapy screening (*Czerniecki et al., 2018*; *Forbes et al., 2018*). The nephron is the functional unit of the kidney, and during differentiation, kidney organoids pattern into early nephron structures with clusters of podocytes and endothelial cells, and a complex tubular epithelial system. Furthermore, organoids show transcriptomic homology to the first trimester human fetal kidney (*Takasato et al., 2015*), and differentiation is further advanced by in vivo implantation (*Bantounas et al., 2018*). Whilst understanding about cell types in kidney organoids has progressed significantly, there is a knowledge gap about extracellular matrix (ECM) and BM assembly during differentiation.

We investigated BM assembly during kidney development using organoids and fetal kidney tissue. With proteomics, we defined a complex sequence of BM assembly during organoid differentiation and demonstrated the utility of this experimental system for investigating BM remodeling in both early development and human disease. Furthermore, we compared by proteomics the organoid matrix to embryonic day (E19) mouse kidney and adult human kidney matrix, and defined the cellular origins of BM components through transcriptomic analyses. Overall, we demonstrate that kidney organoids represent a high-fidelity system to study the dynamics of human BM assembly.

# Results

## Kidney organoids form BM networks that are altered with defective *COL4A5*

To improve the understanding of BM assembly and regulation, we investigated human kidney organoids. We differentiated wild-type iPSCs into intermediate mesoderm cells in 2D culture, and then to 3D kidney organoids (*Figure 1A*, *Figure 1—figure supplement 1A*). We confirmed differentiation to glomerular clusters (WT1[+]/NPHS1[+]/CD31[+]) and CDH1[+] tubular structures in day 18 organoids (*Figure 1B*) and comparable morphology to mouse and human fetal kidney tissues. Day 11 organoids had cell clusters amongst mesenchymal tissue, and at day 14, discernible nephron-like structures (*Figure 1—figure supplement 1A*). At day 18, organoids had regions resembling the nephrogenic zone found at E19 in the mouse and between 8 and 10 weeks post conception (wpc) in human, but lacked distinct corticomedullary differentiation (*Figure 1C*, *Figure 1—figure supplement 1A and B*). By immunofluorescence, we verified the localization of BM integrin receptors adjacent to laminin[+] BM-like structures at day 25 of organoid differentiation (*Figure 1—figure supplement 1C*). Using transmission electron microscopy and immunoelectron microscopy, we observed advanced podocyte differentiation with primary podocyte processes and confirmed assembly of laminin[+] BM structures (*Figure 1D*, *Figure 1—figure supplement 2A and B*). We also detected likely endothelial cells present in glomerular structures (*Figure 1B*, *Figure 1—figure supplement 2B*) and a BM-like matrix between podocytes and endothelial cells in day 25 organoid glomeruli (*Figure 1—figure supplement 2A and B*). Together, these findings demonstrate that kidney organoids mimic the normal progression of kidney differentiation with the concomitant assembly of BM structures in vitro.

To determine the role of organoids as a model to study abnormal BMs in kidney disease, we investigated iPSC lines from patients with Alport syndrome (AS), a genetic disorder caused by variants in collagen IV genes (*Barker et al., 1990*). We selected iPSC lines from a mother and her son, both carrying a likely pathogenic X-linked missense variant in *COL4A5* (c.3695G>A; p.Gly1232Asp) and a variant of unknown significance in *COL4A4* (c.3286C>T; p.Pro1096Ser; *Figure 1E*; see *Supplementary file 1* and Materials and methods 'Clinical presentation' for further details). AS patient-derived organoids progressed through differentiation and formed WT1[+]/NPHS1[+]/CDH1[+] glomeruli and CDH1[+] tubules (*Figure 1F*, *Figure 1—figure supplement 3*) with no evident abnormalities by light microscopy. We found comparable distribution of the collagen IV α4 chain in AS and wild-type organoids (*Figure 1—figure supplement 3*) confirming assembly of a collagen IV α3α4α5 network, which is described in AS patients with missense variants (*Yamamura et al., 2020b*). Since laminin compensation is reported in X-linked AS (*Abrahamson et al., 2007*; *Kashtan et al., 2001*), we examined the deposition of laminin-β2 (LAMB2) in AS organoids. We found increased deposition of LAMB2 in AS organoids, most notable in extraglomerular BM (*Figure 1G*), and further confirmed increased LAMB2 levels by immunoblotting (*Figure 1H*). Together, these findings demonstrate the potential of kidney organoids to reveal abnormal patterns of BM assembly in human development and disease.

## A conserved sequence of BM assembly in kidney organoids

Having identified BM structures in kidney organoids, we next explored the potential for this system to model human BM assembly. Studies in mouse and invertebrate development have shown a sequence for *de novo* BM assembly with initial laminin deposition followed by incorporation of collagen IV, nidogen, and perlecan (*Jayadev et al., 2019*; *Matsubayashi et al., 2017*; *Urbano et al., 2009*). To investigate the assembly sequence in organoids, we used whole-mount immunofluorescence to examine the temporal co-deposition of COL4A1 with laminin (using a pan-laminin antibody) and nidogen with perlecan during differentiation. We found punctate deposits of pan-laminin in interrupted BM networks around cell clusters in day 11 organoids, and in continuous BMs surrounding CD31[+] endothelial and epithelial structures in day 18 and 25 organoids (*Figure 2A*). Conversely, COL4A1 was weakly detected in day 11 organoids but was partially co-distributed with laminin at day 18, and within continuous BM networks at day 25 (*Figure 2A*). Nidogen and perlecan colocalized in discreet, interrupted BMs in day 11 organoids and later in linear BM networks around tubules and NPHS1[+] glomerular structures at days 18 and 25 (*Figure 2B*). Together, these findings indicate that kidney organoids recapitulate the sequence of BM assembly described in vivo, reinforcing their fidelity as system for investigating BM developmental dynamics.

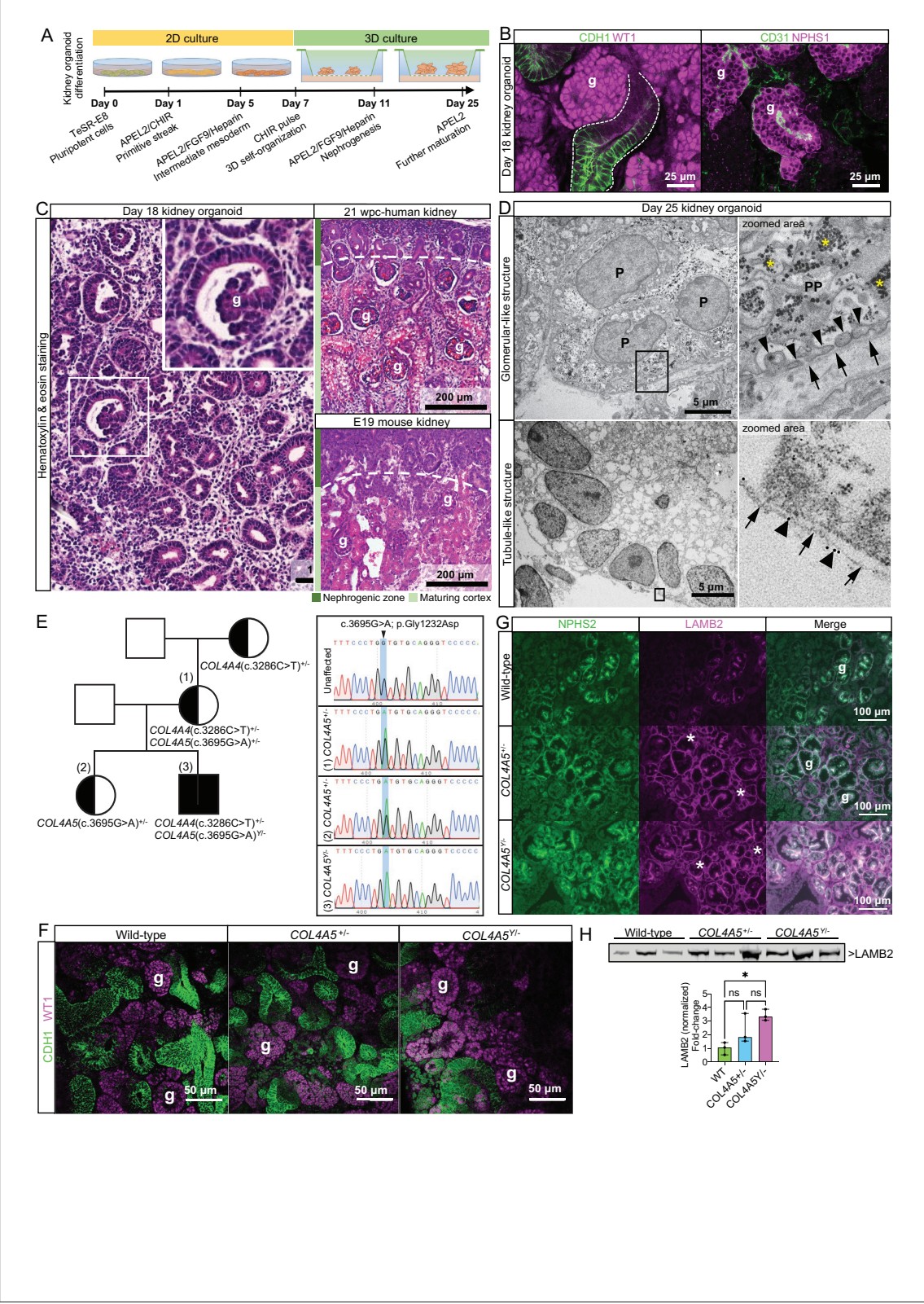

**Figure 1.** Kidney organoid basement membranes are altered in human disease. (**A**) Schematic representing the differentiation of induced pluripotent stem cells (iPSCs) to 3D kidney organoids. (**B**) Whole-mount immunofluorescence for kidney cell types: left image shows glomerular structures (g) with WT1+ cells and CDH1+ tubule segments (dashed line); right image shows a glomerular-like structure (g) containing podocytes (NPHS1+) and endothelial cells (CD31+). (**C**) Representative photomicrographs of day 18 kidney organoids (left) and human and mouse fetal kidneys (right) to demonstrate the

*Figure 1 continued on next page*

*Figure 1 continued*

comparable histological structure; inset shows an organoid glomerular structure (g). (**D**) Transmission electron photomicrographs of glomerular- (upper panels) and tubule-like structures (lower panels) in a day 25 kidney organoid. In the top-right zoomed area, note the features of organoid podocytes (P): a primary process (PP) and distinct intercalating foot processes (thin arrowheads) lining a basement membrane (arrows). Asterisks indicate glycogen granules. In the lower panels, a tubule-like structure in the organoid, and a basement membrane (arrows) labeled with a 10 nm gold-conjugated anti pan-laminin antibody (see large arrowheads in the zoomed area). (**E**) Right: pedigree from a family with a likely pathogenic missense variant in *COL4A5* (c.3695G>A; p.Gly1232Asp, posterior probability 0.988) and an uncertain significance variant in *COL4A4* (c.3286C>T; p.Pro1096Ser [VUS], posterior probability 0.5). Left: Sanger sequencing data for the *COL4A5* variant found in the mother and two siblings, which changes the amino acid from glycine to aspartic acid located in the triple-helical region of the collagen IV trimer. (**F**) Representative whole-mount immunofluorescence images of wild-type and Alport kidney organoids show glomerular structures (g) containing WT1⁺ cells and an intricate cluster of CDH1⁺ epithelial tubules. (**G**) Immunofluorescence for LAMB2 shows increased protein deposition in extraglomerular sites (*). NPHS2 was used as a podocyte marker to identify glomerular structures (g). (**H**) Immunoblotting for LAMB2 using total lysates from wild-type (n = 3) and Alport organoids (n = 3 per group): bar chart shows relative LAMB2 fold change (to wild-type). LAMB2 band optical density was normalized to Ponceau stain (total proteins) and compared by one-way ANOVA and Tukey's multiple comparison tests (*$p$<0.05; ns, not significant). Pooled data are presented as median, and error bars indicate the 95% confidence interval for the median. See *Figure 1—source data 1* (available at https://doi.org/10.6084/m9.figshare.c.5429628). See also *Figure 1—figure supplements 1–3*.

The online version of this article includes the following source data and figure supplement(s) for figure 1:

**Source data 1.** *Figure 1* - source data WB LAMB2.

**Figure supplement 1.** Morphological characteristics of wild-type kidney organoids and human fetal kidney.

**Figure supplement 2.** Ultrastructure of glomerular-like structures in day 25 human kidney organoids.

**Figure supplement 3.** Differentiation of wild-type and Alport kidney organoids.

## Time course proteomics reveals complex dynamics of BM assembly

To understand global BM developmental dynamics, we investigated organoids at days 14, 18, and 25 with time course proteomic analysis. We broadly separated intracellular and extracellular proteins by fractionation (*Figure 3A and B*, *Figure 3—figure supplement 1A*) based on solubility (*Lennon et al., 2014*). Overall, we detected 5,245 proteins in the cellular fraction and 4,703 in the extracellular fraction (*Supplementary file 2*), and by cross-referencing our data with the human matrisome list (*Naba et al., 2016*) we identified 228 matrix proteins in kidney organoids (*Figure 3—figure supplement 1B* and *Supplementary file 2*). Principal component analysis highlighted discreet clustering for the organoid timepoints based on matrix protein abundance (*Figure 3—figure supplement 1C*). There was an increase in abundance of matrix components from day 14 to 25 (*Figure 3B*), and 203 of all matrix proteins found in the organoids (~90%) were detectable at all time points (*Figure 3C*). This initial analysis thus confirmed a gradual assembly of matrix during organoid differentiation. To address global BM composition, we identified BM proteins using the comprehensive BM gene network (*Jayadev et al., 2021*) curated in *basement membrane*BASE (https://bmbasedb.manchester.ac.uk/). The organoid extracellular fraction was enriched for BM proteins compared to the cellular fraction (*Figure 3D*), which was expected as these are large, highly cross-linked proteins, and hence, difficult to solubilize. Furthermore, we observed an increasing trend for BM protein levels through day 14 to 25 (*Figure 3D*), again indicating BM deposition over time and corroborating our immunofluorescence findings. In total, we identified 78 BM proteins (*Figure 3E*), including abundant components that are deposited early in kidney morphogenesis (e.g., COL4A1, COL4A2, LAMA1, LAMB1, LAMC1) (*Figure 3F*, *Figure 3—figure supplement 1D*). LAMA5 and LAMB2, two key components of the mature GBM, only appeared amongst the most abundant BM components at day 25 (*Figure 3—figure supplement 1D*), thus indicating a temporal expression of GBM laminins during organoid differentiation (*Figure 3F*). This was confirmed by marked upregulation of mature GBM proteins from day 18 to day 25, with LAMB2 scoring with the highest fold change followed by LAMA5 and COL4A3 (*Figure 3G and H*). LAMA5 was also enriched from day 14 to day 18 together with other GBM proteins (COL4A3, AGRN) and early BM collagens and laminins (COL4A1, COL4A2, LAMC1, LAMA1) (*Figure 3G and H*, *Figure 3—figure supplement 1E*).

During GBM assembly, an initial laminin-111 (α1αβ1γ1) network is sequentially replaced by laminin-511 then -521 (*Abrahamson et al., 2013*). We therefore reasoned that day 14 to day 18 would represent a period of intense BM assembly and initial GBM differentiation. In support of this hypothesis, a pathway enrichment analysis of the proteins upregulated from day 14 to 18 revealed an over-representation of terms associated with BM assembly and remodeling, including laminin interactions,

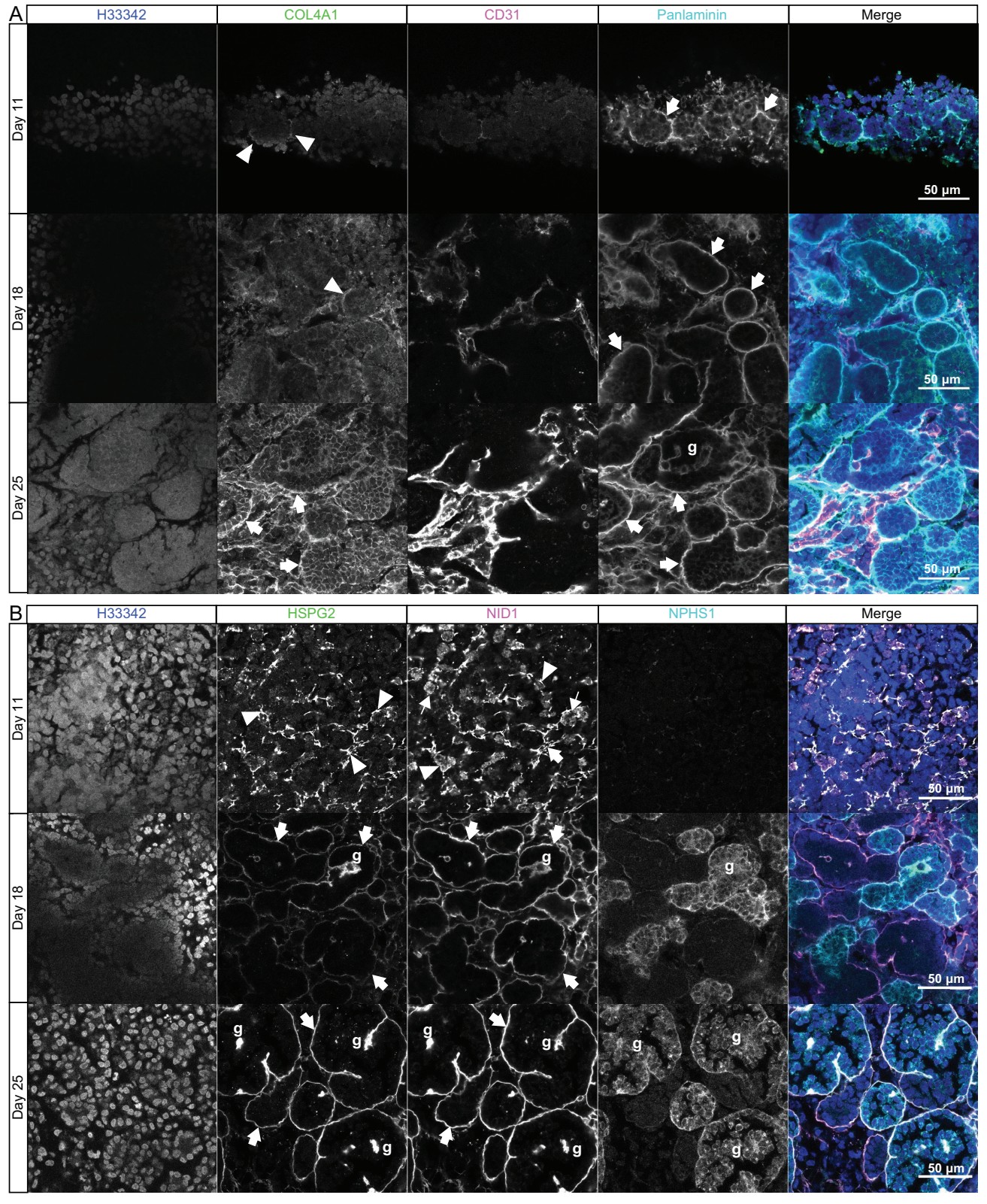

**Figure 2.** Sequential assembly of basement membrane components. (**A**) Confocal immunofluorescence microscopy of wild-type kidney organoids showing the temporal emergence and co-distribution of COL4A1 and pan-laminin, and (**B**) perlecan and nidogen at days 11, 18, and 25 of differentiation. NPHS1 and CD31 were used as markers for podocyte and endothelial cells, respectively, in glomerular-like structures (g). Arrowheads indicate interrupted BM segments, large arrows indicate diffuse BM networks, and thin arrows indicate intracellular droplets of proteins.

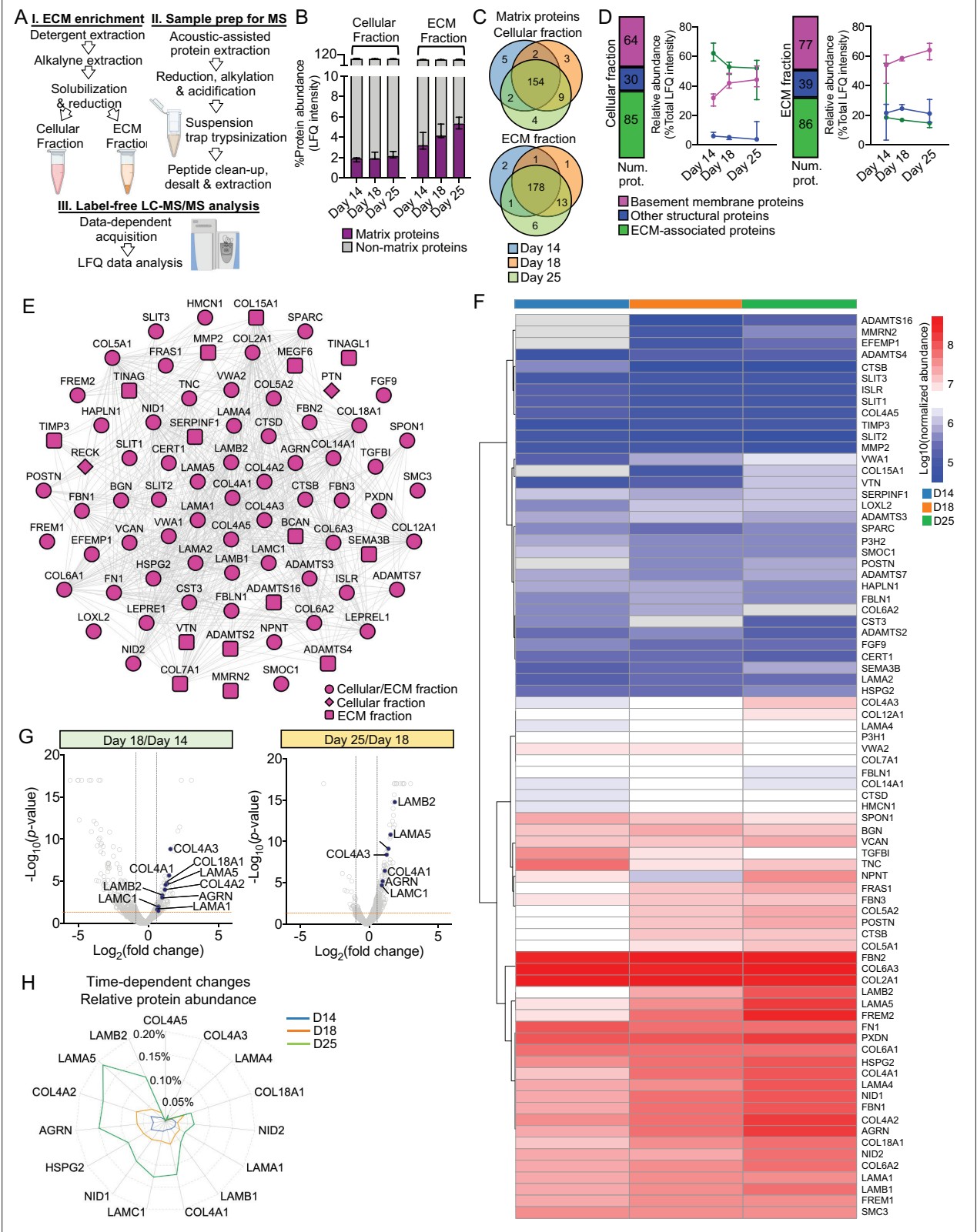

**Figure 3.** Time course proteomics reveals complex dynamics of basement membrane assembly. (**A**) Schematic of sample enrichment for matrix (ECM) proteins for tandem mass spectrometry (MS) analysis (created with BioRender.com). (**B**) Bar charts show the relative abundance of matrix and non-matrix proteins identified by MS analysis in the cellular and ECM fractions of kidney organoids at days 14, 18, and 25 (n = 3 pools per time point). Pooled data are presented as median, and error bars indicate the 95% confidence interval for the median. (**C**) Venn diagrams showi the identification overlap for

*Figure 3 continued on next page*

*Figure 3 continued*

matrix proteins detected in kidney organoids at days 14, 18, and 25. (**D**) Matrix proteins are classified as basement membrane, other structural and ECM-associated proteins. Bar charts show the number of matrix proteins per matrix category in both cellular and ECM fractions, and line charts show the changes in their relative abundance (percentage of total matrix abundance) over the time course differentiation. Pooled data are presented as median, and error bars indicate the 95% confidence interval for the median. (**E**) Protein interaction network showing all basement membrane proteins identified over the kidney organoid time course MS study (nodes represent proteins and connecting lines indicate reported protein-protein interactions). (**F**) Heat map showing the $\log_{10}$-transformed abundance levels of basement membrane proteins identified in the ECM fraction along kidney organoid differentiation time course (proteins detected only at one time point are not shown). (**G**) Volcano plots show the $\log_2$-fold change (x-axis) versus $-\log_{10}$-p-value (y-axis) for proteins differentially expressed in the ECM fraction of kidney organoids from day 14 to day 18, and from day 18 to day 25 (n = 3 per time point). Key basement membrane proteins significantly upregulated (FC $\geq$ 1.5, p-value<0.05, two-way ANOVA test, n = 3) are indicated. (**H**) Time-dependent changes in the relative abundance (percentage of total protein intensity) of key basement membrane proteins in the ECM fraction of kidney organoids during differentiation. Pooled data are presented as median. See *Figure 3—figure supplement 1*.

The online version of this article includes the following figure supplement(s) for figure 3:

**Figure supplement 1.** Time course proteomic analysis of kidney organoid differentiation.

degradation of ECM and collagen chain trimerization (*Figure 3—figure supplement 1F*). Together, this global proteomic analysis revealed new insights into the complexities of BM dynamics during kidney organoid differentiation and the distinct temporal emergence of BM isoforms required for long-term functional integrity of the GBM.

## Tracking collagen IV and laminin isoforms during kidney organoid differentiation

To confirm the temporal sequence of specific BM isoforms, we investigated the distribution of COL4A1, COL4A3, LAMA5, LAMB1 and LAMB2 in kidney organoid BMs at days 14, 18 and 25 by immuno-fluorescence (*Figure 4A*). As described earlier, COL4A1 appeared at day 11 (*Figure 2A*) and later partially colocalized with laminin at day 14 and as continuous BM networks from day 18 (*Figure 4A*). Conversely, COL4A3 was scarce at days 14 and 18, but clearly colocalized with laminin in glomerular structures at day 25 (*Figure 4A*). We detected LAMA5 from day 18, and this increased in glomerular structures at day 25. LAMB1 was widely distributed from day 14 to day 25, whereas LAMB2, detected from day 18 onwards, was enriched in glomerular structures at day 25. These findings not only confirm a temporal emergence of specific BM isoforms in kidney organoids, but also highlight their specific localization to glomerular structures later in differentiation.

We then hypothesized that distinct cell types would express specific BM isoforms to concentrate their distribution and therefore reanalyzed publicly available day 25 kidney organoids single-cell RNA sequencing (scRNA-seq) data (*Combes et al., 2019b*) to map the expression profile for BM genes (*Figure 4—figure supplement 1A*, *Supplementary file 3*). We found that NPHS2[+]/PODXL[+] podocytes were the main source of *COL4A3* and *COL4A4* (*Figure 4B*, *Figure 4—figure supplement 1A*), and they also had high levels of expression for *LAMA5* and *LAMB2*. PECAM1[+]/KDR[+] endothelial cells, MAB21L2[+]/CXCL14[+] stromal cells, and PAX8[+]/PAX2[+] nephron cell lineages all expressed *COL4A1*, *COL4A2*, *LAMB1* and *LAMC1*. *LAMA1* was detected in the nephron cell cluster, whereas LAMA4 was expressed by both endothelial and stroma cells (*Figure 4B*, *Figure 4—figure supplement 1A*). These findings align with current understanding of kidney development in vivo and indicate that kidney organoids recapitulate the known cell-specific contributions to BM assembly during glomerulogenesis.

Since the developmental transition from the α1α1α2 to the α3α4α5 network of collagen IV is key for long-term GBM function and reduced or absent α3α4α5 leads to loss of GBM function (*Miner and Sanes, 1996*), we mapped the localization of collagen IV isoforms in day 25 organoids and compared to E19 mouse glomeruli (*Supplementary file 4*). With proteomic analysis, we identified 25 BM proteins in laser-microdissected maturing E19 glomeruli, and these were also detected in kidney organoids. Identifications included COL4A1, COL4A2, and COL4A3 (*Figure 4C*), thus indicating the presence of both α1α1α2 and α3α4α5 networks. We compared the localization of COL4A1 and COL4A3 by immunofluorescence in NPHS2[+] glomeruli in mouse and kidney organoids (*Figure 4D*). We observed a similar distribution of COL4A1, but for COL4A3, we found GBM-like and extraglomerular distribution in kidney organoids (*Figure 4D*). Therefore, kidney organoids initiate collagen IV isoform transitions during glomerulogenesis. To further explore mechanisms of isoform transitions, we investigated the expression of LIM homeodomain transcription factor 1-beta (LMX1b) and FERM-domain protein

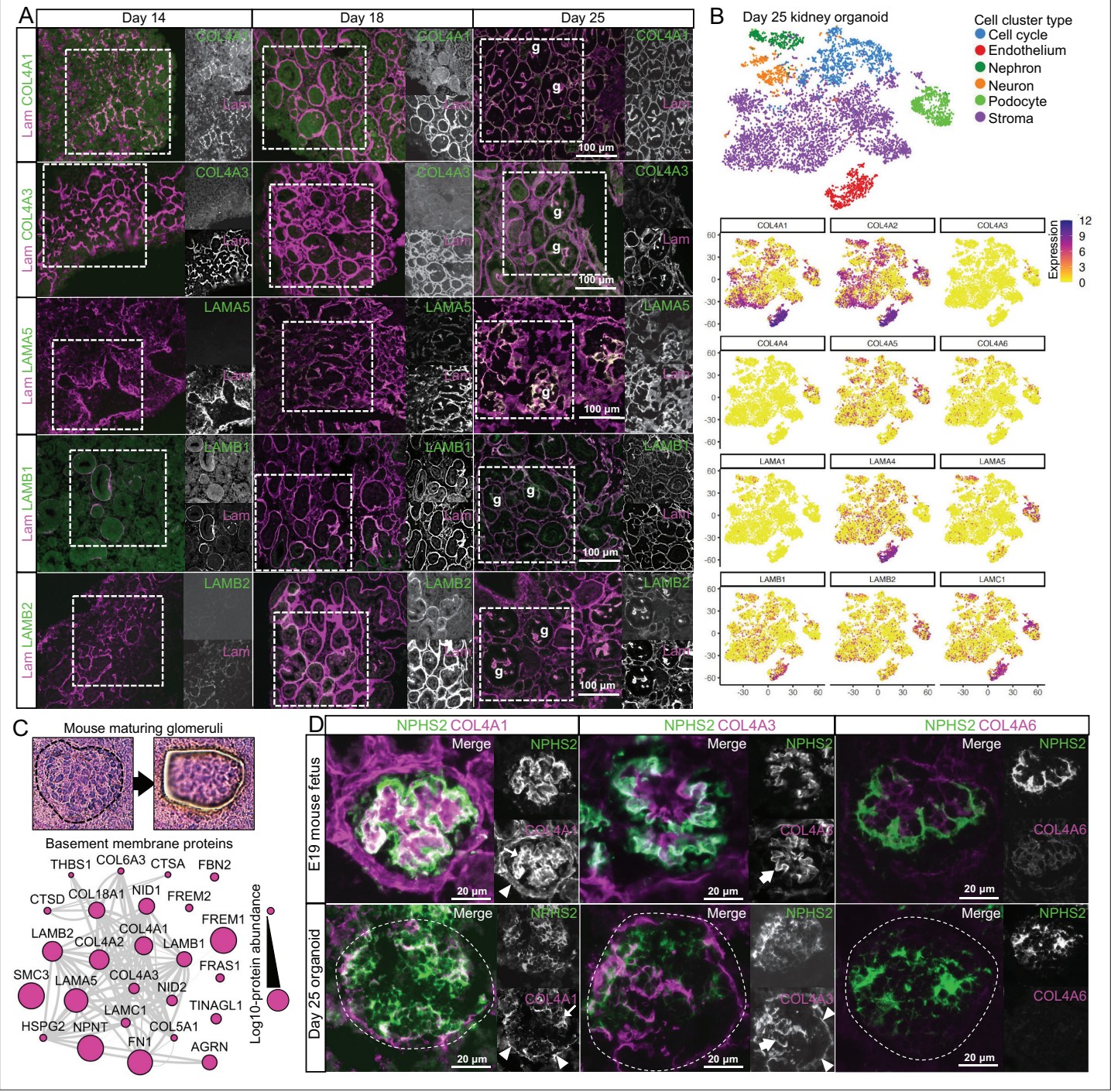

**Figure 4.** Key collagen IV and laminin isoform transitions occur during kidney organoid differentiation. (**A**) Immunofluorescence for key collagen IV and laminin isoforms shows their emergence and distribution in kidney organoid basement membrane. Pan-laminin antibody was used to co-label organoid basement membranes; glomerular structures are indicated (g). (**B**) Reanalysis of a kidney organoid scRNA-seq dataset GSE114802 (***Combes et al., 2019b***) confirms cellular specificity for collagen IV and laminin isoform gene expression. tSNE plots represent the cell-type clusters identified, and color intensity indicates cell-specific level of expression for selected basement membrane genes. (**C**) Proteomic analysis of laser-captured maturing glomeruli from E19 mouse kidneys (n = 4). Histological images show a laser-microdissected glomerulus, and the protein interaction network shows the 25 basement membrane proteins identified (nodes represent proteins and connecting lines indicate reported protein-protein interactions). (**D**) Immunofluorescence for specific collagen IV isoforms in maturing glomeruli in E19 mouse kidney and in glomerular structures (indicated by dashed lines) in day 25 organoids. NPHS2 was used to label podocytes. Arrowheads indicate the Bowman's capsule in the mouse or the glomerular surface in the organoid; large arrows indicate the GBM in the mouse or GBM-like matrix in the organoid; thin arrows indicate mesangial matrix in the mouse or

*Figure 4 continued on next page*

*Figure 4 continued*

internal glomerular matrix deposition in the organoid. See *Figure 4—figure supplement 1*.

The online version of this article includes the following figure supplement(s) for figure 4:

**Figure supplement 1.** Single-cell RNA-sequencing data analysis of human kidney organoids.

EPB41L5 in day 25 organoids (*Figure 4—figure supplement 1B*). LMX1b and EPB41L5 are proposed regulators of GBM assembly and isoform transitions during development (*Maier et al., 2021*; *Morello et al., 2001*). Moreover, EPB41L5 is also implicated in regulating the incorporation of laminin-511 and -521 into stable GBM scaffolds (*Maier et al., 2021*). We found that similar cell populations expressed *EPB41L5* and *LAMA5* (*Figure 4—figure supplement 1B*), and in the proteomic analysis there was an increase in EPB41L5 protein levels from day 14 to day 18 (*Figure 4—figure supplement 1C*) coinciding with an increase in LAMA5 protein levels (*Figure 3G and H*). Together, these findings demonstrate that kidney organoids initiate isoform transitions during glomerular differentiation with the expression of known BM regulators.

## BMs in late-stage organoids and fetal kidneys are highly correlated

To relate BM assembly in kidney organoids to a comparable in vivo system, we examined whole fetal human and mouse kidneys. Having verified morphological similarity between day 25 organoids and E19 mouse kidney (*Figure 1C*), we used this mouse and organoid time points for comparison by proteomic analysis. We generated cellular and extracellular fractions from whole E19 mouse kidneys (*Figure 5A*, *Figure 5—figure supplement 1A and B*) and identified 208 matrix components from a total of 5,071 proteins (*Figure 5A*, *Supplementary file 5*). These included 83 BM proteins, and the most abundant were those seen early (COL4A1, COL4A2, COL18A1, LAMB1, LAMC1) and later (LAMA5) in kidney BM assembly (*Figure 5—figure supplement 1C*). As with kidney organoids, we found an enrichment for BM proteins in the ECM fraction obtained from the mouse fetal kidney (*Figure 5B*). We next compared these findings to an E18.5 mouse kidney proteomic dataset (*Lipp et al., 2021*) and found considerable identification overlap, with a further 130 matrix protein detected in our study (*Figure 5C*), including key GBM components COL4A3, COL4A4 and COL4A5, and also COL4A6. We then compared the BM proteins detected in each organoid time point with the E19 mouse kidneys (*Figure 5D*) and found the highest overlap (58.4%) between E19 and day 25 kidney organoids. In line with previous findings (*Hale et al., 2018*), this comparison highlighted later expression of TINAG and TINAGL1 in day 25 organoids, also detected in the E19 mouse kidneys but not in day 14 or day 18 organoids. TINAGL1 was also detected in laser-microdissected E19 mature glomeruli (*Figure 4C*) and in proteomic studies of adult human glomeruli (*Lennon et al., 2014*), but its role in BM biology remains unknown. In addition, the overlap of other structural and matrix-associated proteins was lower than for BM proteins (*Figure 5—figure supplement 1D*), which highlights conservation of BM composition between mouse and kidney organoids. To further verify similarities, we performed a Spearman's rank correlation and found that E19 mouse kidneys had the higher correlation with more differentiated organoids (i.e., days 18 and 25; *Figure 5E*, *Figure 5—figure supplement 1E*).

Next we analyzed an E18.5 mouse kidney scRNA-seq dataset (*Combes et al., 2019a*; *Figure 5F*) and from 8/9 wpc human kidneys (*Young et al., 2018*; *Figure 5—figure supplement 2*) to identify cells expressing specific BM genes (*Supplementary file 3*). In the mouse, we found mature GBM components expressed by *Synpo*[+]/*Nphs2*[+] podocytes (*Col4a3*, *Col4a4*, *Lama5*, *Lamb2*), *Plvap*[+]/*Pecam1*[+] vascular cells (*Lama5*, *Lamb2*), and *Cited1*[+]/*Crym*[+] nephron progenitor cell lineages, and *Aldob*[+]/*Fxyd2*[+] tubular, vascular, and *Six2*[+] stromal cells predominantly contributing to *Col4a1*, *Col4a2*, *Lamb1*, and *Lamc1* expression (*Figure 5F*). *Lama1* was mainly expressed by *Clu*[+]/*Osr2*[+] S-shaped bodies and *Gata3*[+]/*Wfdc2*[+] ureteric bud/distal tubular cells, *whereas Lama4* expression was restricted to vascular and stromal cells. A similar pattern was observed in the embryonic human kidney (*Figure 5—figure supplement 2*), and these findings were also consistent with our findings in day 25 kidney organoids (*Figure 4B*, *Figure 4—figure supplement 1*). Interestingly, immune cells also contributed to *Col4a1* and *Col4a2* expression in both mouse and human kidneys. Collectively, these findings highlight the conservation of BM gene expression across organoids and human and mouse fetal kidneys.

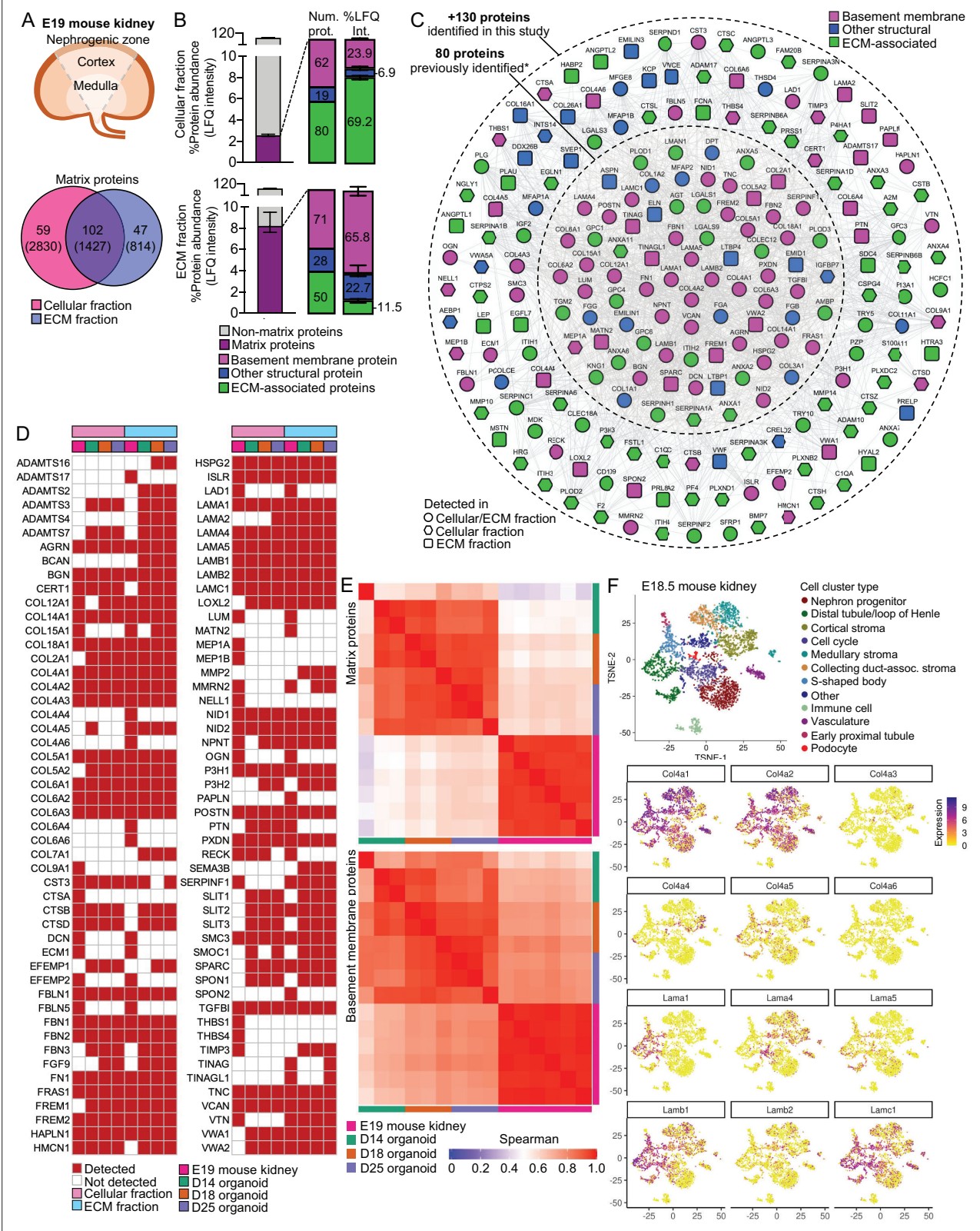

**Figure 5.** Basement membranes in mouse fetal kidneys are comparable to kidney organoids. (**A**) Schematic representation of the E19 mouse kidney sampled for mass spectrometry (MS)-based proteomics, and a Venn diagram showing the overlap for matrix proteins identified in the cellular and extracellular matrix (ECM) fractions. (**B**) Bar charts show enrichment levels for matrix proteins in both cellular and ECM fractions (n = 6), as indicated by the number and relative abundance of proteins in each matrix category. Pooled data are presented as median, and error bars indicate the 95%

*Figure 5 continued on next page*

*Figure 5 continued*

confidence interval for the median. (**C**) Expanded mouse fetal kidney matrisome represented as a protein interaction network (nodes represent proteins identified in this and in a previous study [*Lipp et al., 2021*], and connecting lines indicate reported protein-protein interactions). (**D**) Comparison of basement membrane proteins identified in the E19 mouse kidney* (MFK) and human kidney organoids (HKOs) during differentiation (asterisks indicate that corresponding human ortholog for mouse proteins are shown). (**E**) Spearman's rank correlation analysis of matrix and basement membrane protein abundance (in the ECM fraction) comparisons between E19 MFK and human kidney organoids (HKO) at days 14 (D14), 18 (D18) and 25 (25D). (**F**) Reanalysis of an E18.5 mouse kidney scRNA-seq dataset GSE108291 (*Combes et al., 2019a*) confirms cellular specificity for collagen IV and laminin isoform gene expression. tSNE plots represent the cell-type clusters identified, and color intensity indicates cell-specific level of expression for selected basement membrane genes. See *Figure 5—figure supplements 1 and 2*.

The online version of this article includes the following figure supplement(s) for figure 5:

**Figure supplement 1.** Proteomic analysis of embryonic day 19 (E19) mouse kidney and correlational comparison with kidney organoid proteomics.

**Figure supplement 2.** Single-cell RNA-sequencing analysis of human fetal kidney.

## BMs are dynamic through embryonic development to adulthood

Having observed dynamic BM composition during kidney development, we then compared to composition in adulthood. For this, we analyzed proteomic data from isolated adult human nephron compartments and identified 71 BM proteins in the glomerulus, and 61 in the tubulointerstitium (*Supplementary file 6*). We compared BM networks in the adult kidney with both developmental systems (day 25 kidney organoids and E19 mouse kidney) and found a significant identification overlap with 44 of 107 BM proteins shared amongst all datasets. These included core components (COL4A1, COL4A2, LAMA1, LAMB1, LAMC1, HSPG2, COL18A1, NID1, NID2), mature GBM components (COL4A3, COL4A5, LAMA5, LAMB2), and many minor structural proteins (*Figure 6A*). We found a strong rankcorrelation between both matrix and BM networks in adult human glomerular and tubulointerstitial compartments, but lower correlation between adult and developmental datasets (*Figure 6B*, *Figure 6—figure supplement 1*). Interestingly, the rankcorrelation between adult and development data was lower for BM than for all matrix proteins, suggesting that although kidney BMs retain a consistent profile through development to adulthood, there is diversification within distinct kidney compartments.

To understand the cellular origins of BM components through development to adulthood, we reanalyzed an adult human kidney scRNA-seq dataset (*Young et al., 2018*; *Supplementary file 3*), and as with organoids and mouse fetal kidney, we found *COL4A1*, *COL4A2*, *LAMB1*, and *LAMC1* to be predominantly expressed by endothelial and tubular cells. Although podocyte markers were not enriched in the adult dataset, we detected *COL4A3* and *COL4A4* expression by $PAX8^+$ nephron cell types, and *LAMA5* and *LAMB2* mainly expressed by $KRT5^+/EMCN^+$ endothelial cells (*Figure 6C*). *LAMA4* was widely detected in both E18.5 mouse kidney and day 25 organoids (*Figures 3B and 4F*, *Figure 4—figure supplement 1*), but barely present in the adult kidney (*Figure 6C*), consistent with previous reports of transient expression in human kidney development (*Miner, 1999*). These findings demonstrate consistent cellular origins for BM components through development to adulthood.

To further verify the extent of this consistency, we selected minor BM components across all scRNA-seq datasets (*Figure 6D*). These included *FBLN1* and *TGFBI* both implicated in BM remodeling (*Boutboul et al., 2006*; *Feitosa et al., 2012*); *FRAS1* and *FREM2* important for branching morphogenesis (*Chiotaki et al., 2007*; *Petrou et al., 2008*); and *TINAG* and *TINAGL1*, with unknown roles in BM function. We found a common pattern of enrichment amongst the developmental datasets for *FBLN1*, which was expressed by stromal cells; *FRAS1* and *FREM2* were expressed by podocyte and nephron cell clusters; and *TGFBI* was expressed by stromal cells, nephron progenitors, and immune cells. In human adult kidneys, *FBLN1* was mainly present in pelvic epithelial cells and *FRAS1/FREM2* in ureteric bud/distal tubule cells, thus indicating spatiotemporal expression of these components. *TINAG* and *TINGAL1* had variable patterns of cell expression across datasets. This comparative analysis shows the conservation of distinct cell types contributing to BM assembly during kidney development and uncovers the diversification in cellular origins of BM components in adult kidneys.

## Discussion

Mammalian kidney development involves a series of morphogenetic events that proceed in an orchestrated manner to give rise to ~1,000,000 nephrons in the human kidney and 1000s in the mouse. Many

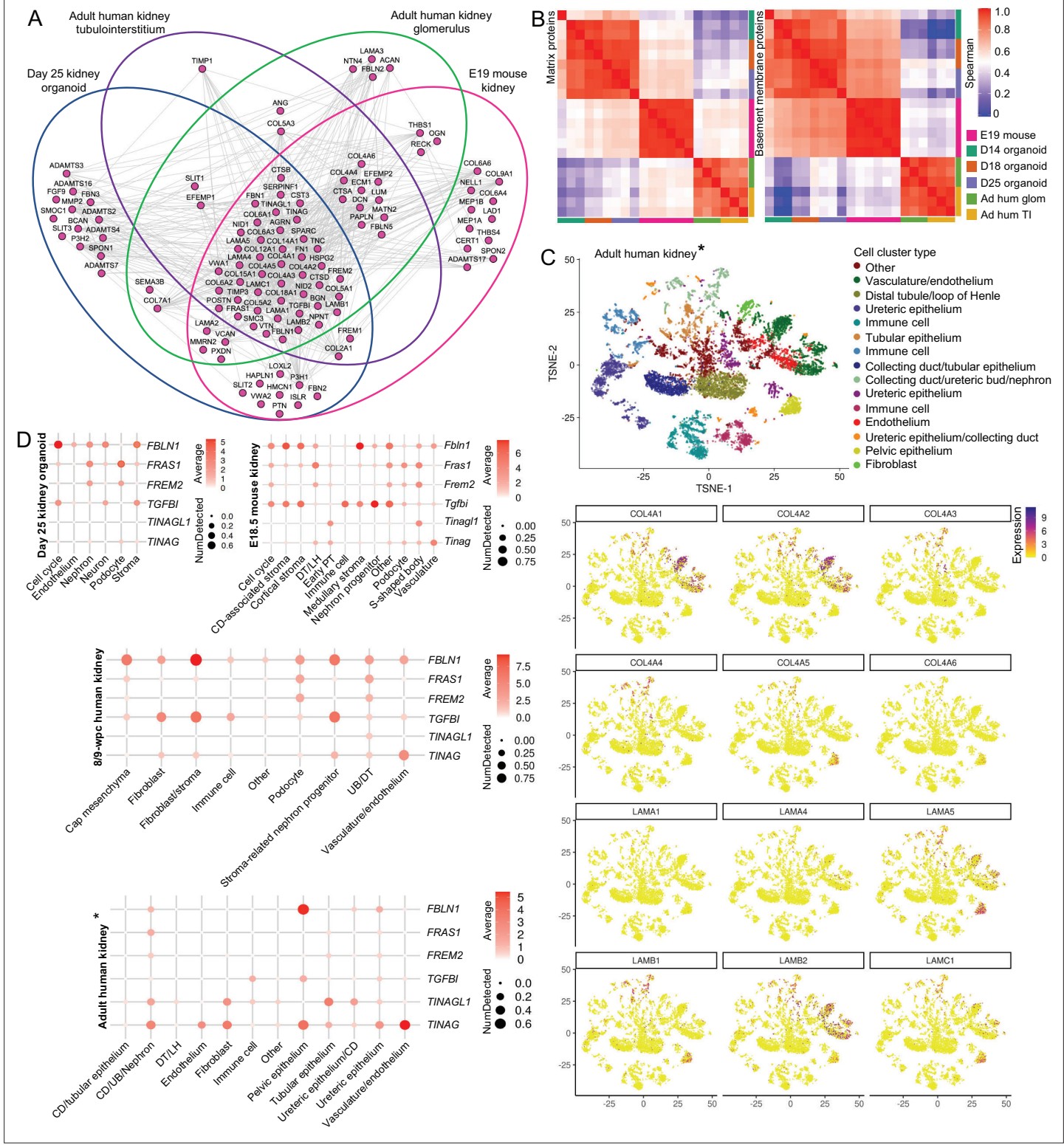

**Figure 6.** Basement membranes are dynamic through embryonic development to adulthood. (**A**) Integrative interactome shows a common core of 44 basement membrane proteins across day 25 organoid, E19 mouse kidney, and adult human glomerular and tubulointerstitial compartments. Venn diagrams indicate in which dataset each BM protein was detected. Nodes represent proteins, and lines indicate reported protein-protein interactions. (**B**) Spearman's rank correlation analysis of matrix and basement membrane protein abundance (in ECM fraction) comparisons between E19 mouse fetal kidney (MFK), human kidney organoids (HKO), and adult human glomerulus and kidney tubulointerstitium. (**C**) Reanalysis of an adult human kidney scRNA-seq dataset EGAS00001002553 (*Young et al., 2018*) confirms cellular specificity for collagen IV and laminin isoform gene expression. tSNE plots

*Figure 6 continued on next page*

*Figure 6 continued*

represent the cell-type clusters identified, and color intensity indicates cell-specific level of expression for the selected basement membrane genes (*proximal tubule cells were not included). (**D**) Cell expression of minor basement membrane components through kidney development to adulthood. Dot plots show the level of expression of target genes in all publicly available datasets reanalyzed in this study (*proximal tubule cells were not included). See *Figure 6—figure supplement 1*.

The online version of this article includes the following figure supplement(s) for figure 6:

**Figure supplement 1.** Integrated correlational analysis of organoid and in vivo kidney datasets.

of these processes require spatiotemporal assembly and remodeling of BMs throughout nephrogenesis (*Figure 7*). Here, we demonstrate the fidelity of human kidney organoids as a system for investigating assembly and regulation of kidney BMs in health and disease with the following key findings: (1) the identification of a conserved sequence of BM component assembly during kidney organoid differentiation, (2) evidence of global BM dynamics during organoid differentiation with high correlation to fetal kidney BM composition, and (3) the diversification of the cellular origin of BM components during kidney development.

BMs are complex structures, and proteomic studies have highlighted this complexity in homeostasis and disease (*Lennon et al., 2014*; *Randles et al., 2015*). During development, BMs are also highly dynamic and undergo intense remodeling (*Kyprianou et al., 2020*), but understanding of BM assembly and regulation is limited by the lack of appropriate systems to track components that are spatiotemporally regulated and undergo turnover (*Naylor et al., 2021*). Human fetal tissue has limited availability and developmental studies are restricted to static time points, and the technical limitations of imaging in mouse models also difficult the investigation of the dynamic BM environment in vivo. However, BM studies in *Drosophila* and *C. elegans* have provided important insights into BM dynamics and turnover using fluorescent tagging of endogenous proteins (*Keeley et al., 2020*; *Matsubayashi et al., 2020*), and these studies highlight the power of studying BM dynamics.

Despite the developmental limitations with iPSC-derived kidney organoids, including lack of features such as directional cues, vascularization, and cortical-medullary segmentation (*Romero-Guevara et al., 2020*), this system has morphological and molecular features comparable to fetal kidney tissue that overcome species differences and provides a complex in vitro environment to examine BM assembly (*Bantounas et al., 2018*). We demonstrated that kidney organoids differentiate into glomerular structures containing the cells required for GBM assembly, andother single-cell transcriptomic studies have also shown over 20 other distinct cell populations present in kidney organoids (*Combes et al., 2019a*; *Wu et al., 2018*). Cross-talk between different cell types is essential for BM assembly and influences its composition (*Byron et al., 2014*), hence, the multiple cell types in the organoid system enable BM formation and remodeling. A key finding from this study is that organoids form BMs early during differentiation and, more importantly, recapitulate a sequence of BM assembly events with initial deposition of laminin followed by incorporation of collagen IV, nidogen, and perlecan (*Brown et al., 2017*; *Sasaki et al., 2004*).

Kidney organoids have also provided new insights into disease processes (*Rooney et al., 2021*; *Tanigawa et al., 2018*; *Tian and Lennon, 2019*). We found that kidney organoids with a pathological missense variant in *COL4A5* differentiated and deposited core BM proteins, including a collagen IV α3α4α5 network, which is described in missense variant cases. In one study, 64 out of 146 patients with X-linked AS had the collagen IV α5 chain in the GBM (*Yamamura et al., 2020a*). Despite evidence of protein secretion, the GBM fails to maintain function in these patients. Interestingly, we found increased deposition of LAMB2 in extraglomerular BMs in our Alport organoids. Dysregulation of glomerular laminins, including LAMB2, in patients with AS and animal models has been reported (*Abrahamson et al., 2007*; *Kashtan et al., 2001*), but the mechanisms for this are unclear. Our findings in Alport organoids demonstrate the utility of this system to dissect abnormal mechanisms of BM assembly.

To define global BM composition in kidney organoids, we used MS-based proteomics and identified 78 BM proteins dynamically expressed throughout the kidney organoid differentiation time course. Core GBM components including laminin-521 and collagen IV α3α4α5, which only appear in mature glomeruli, were also detected. Developmental isoform transitions in the GBM involving laminin and collagen IV are described in humans and rodents (*Abrahamson and St John, 1993*), and in the current understanding, immature glomeruli assemble a primary GBM containing laminin-111

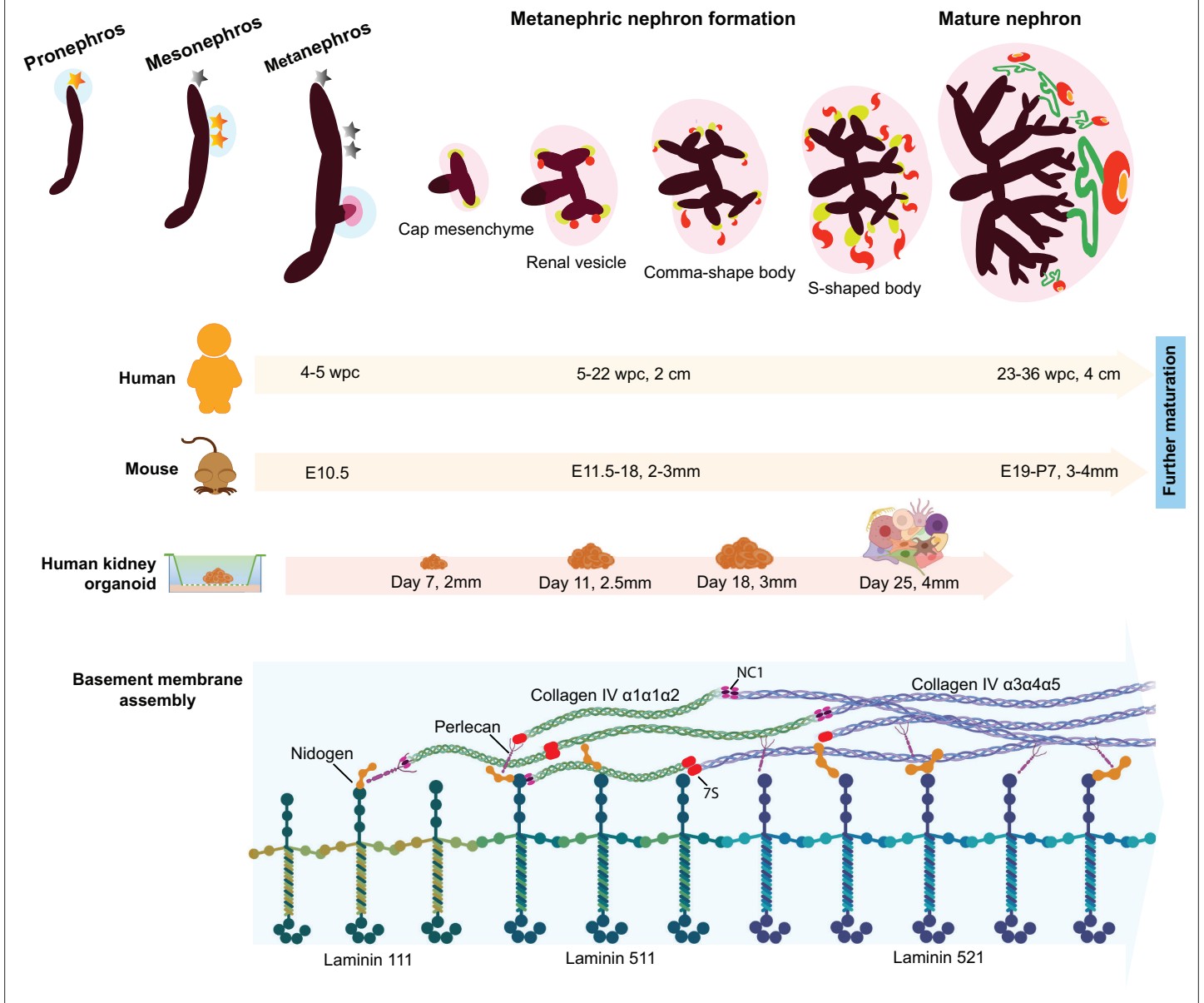

**Figure 7.** Overview of the developmental milestones in human and mouse kidney morphogenesis and a comparison to human kidney organoids. Differentiation is dated in in weeks post conception (wpc) for human, embryonic days (E) for mouse, and in days of differentiation for kidney organoids; and measures (cm or mm) indicate specimen size in each model. Three sets of embryonic kidneys develop in mammals in a temporal sequence: from the pronephros to the mesonephros (both temporary), and then to the metanephros (permanent). Metanephric nephron formation (nephrogenesis) commences in humans at 5 wpc, in mice at E10.5, and in kidney organoid from day 11, when laminin starts to deposit as basement membrane networks. Following the same developmental stage in human and mouse kidneys, kidney organoids start to form discernible renal vesicles at day 14, and distinct comma- and S-shaped bodies at day 18; day 25 organoids, which most closely resemble late capillary loop stage nephrons in vivo, comprise more mature structures including glomeruli with capillary lumens, proximal tubules, and distal tubules. Meanwhile, a conserved sequence of basement membrane assembly is detected in kidney organoids, and laminin and collagen IV developmental isoform transitions are identified between day 14 and day 25.

and collagen IV α1α2α1 that is later replaced, respectively, by laminin-521 and collagen IV α3α4α5 in mature glomeruli. In keeping with these observations, we found a temporal and spatial emergence of mature collagen IV and laminin isoforms within glomerular structures in kidney organoids, and in a reanalysis of publicly available scRNA-seq data, we confirmed podocyte expression of mature GBM markers. These findings demonstrate that kidney organoids can efficiently recapitulate the spatiotemporal emergence of GBM components. The triggers for GBM isoform switching remain unknown, but

two regulators, LMX1b and EPB41L5, have been implicated in this. Studies in *Lmxb1* knockout mice suggest that podocyte expression of this transcription factor is not essential for initial GBM assembly but is linked to *Col4a3/Col4a4* expression during glomerulogenesis as demonstrated by reduced collagen IV α3α4α5 network in the GBM in null *Lmxb1* newborn mice (*Morello et al., 2001*). We found podocyte-specific expression of *LMX1b/COL4A3/COL4A4* in day 25 kidney organoids and, moreover, confirmed deposition of COL4A3/COL4A4 in a GBM pattern within NPHS2⁺-glomerular structures in these organoids. In addition, podocyte expression of EPB41L5, a component of the podocyte integrin adhesion complex, was linked to GBM assembly in vivo and incorporation of laminin-511/521 into podocyte-derived extracellular BM networks in vitro (*Maier et al., 2021*). In this study, we detected EPB41L5 in kidney organoid and mouse proteomic datasets, with cell expression patterns matching that of *LAMA5*, indicating the potential for this system to unravel further insights into GBM regulation.

There are a few proteomic studies addressing the spatiotemporal changes in BM during development. One study of mouse kidney development defined the global ECM composition through development to adulthood and described a sequence of changes in the interstitial matrix over development (*Lipp et al., 2021*). With our sample fractionation strategy and analytical pipeline, we detected a further 130 matrix proteins, including key GBM components (COL4A3, COL4A4, COL4A5) and BM regulators such as PAPLN and HMCN1 (*Keeley et al., 2020*; *Morrissey et al., 2014*). Our findings further demonstrated that late-stage kidney organoids (at day 25) and mouse kidneys at E19 share a very comparable BM protein profile. We also verified similar patterns of BM gene expression between kidney organoids and human and mouse fetal kidneys, especially for collagen IV and laminin isoforms. Overall, these data demonstrate the high fidelity with which kidney organoids recapitulate BM gene expression and protein composition seen in vivo. Thus, we conclude that kidney organoids are a highly tractable model that can be used to study the dynamic nature of human BM assembly in both health and disease.

# Materials and methods

### Key resources table

| Reagent type (species) or resource | Designation | Source or reference | Identifiers | Additional information |
|---|---|---|---|---|
| Strain, strain background (*Mus musculus*) | *Swiss* | University of São Paulo (Brazil) | - | 2 months old, male and female mice |
| Cell line (*Homo sapiens*) | iPSC | HipSci | CGAP-38728; CGAP-4852B; CGAP-581E8 | Derived from patients with Alport syndrome |
| Cell line (*Homo sapiens*) | iPSC | *Wood et al., 2020* | - | Derived from peripheral blood mononuclear cells |
| Biological sample (*Homo sapiens*) | Embryonic and fetal kidneys | Joint MCR/Wellcome Trust HDBR | Kidney_ID: 13878; 11295; 13891; 13842; 1358 | FFPE samples |
| Antibody | Anti-CD31, clone 89C2 (mouse monoclonal) | Cell Signaling | Cat# 3582 | IF (1:100) WM (1:100) |
| Antibody | Anti-E-cadherin, clone M168 (mouse monoclonal) | Abcam | RRID:AB_1310159 | IF (1:300) WM (1:300) |
| Antibody | Anti-WT1, clone C-19 (rabbit polyclonal) | Santa Cruz Biotechnology | RRID:AB_632611 | IF (1:100) WM (1:100) |
| Antibody | Anti-human nephrin (sheep polyclonal) | R&D Systems | RRID:AB_2154851 | IF (1:200) WM (1:400) |
| Antibody | Anti-human collagen IV α1 chain NC1 domain, clone H11 (rat monoclonal) | Chondrex | Cat# 7070 | IF (1:100) WM (1:400) |
| Antibody | Anti-human collagen IV α3 chain NC1 domain, clone H31 (rat monoclonal) | Chondrex | Cat# 7076 | IF (1:100) |
| Antibody | Anti-human collagen IV α4 chain NC1 domain, clone H43 (rat monoclonal) | Chondrex | Cat# 7073 | IF (1:100) |
| Antibody | Anti-human collagen IV α6 chain NC1 domain, clone H63 (rat monoclonal) | Chondrex | Cat# 7074 | IF (1:50) |

*Continued on next page*

*Continued*

| Reagent type (species) or resource | Designation | Source or reference | Identifiers | Additional information |
|---|---|---|---|---|
| Antibody | Anti-laminin (rabbit polyclonal) | Abcam | RRID:AB_298179 | IF (1:250) WM (1:250) |
| Antibody | Anti-nidogen, clone 302,117 (mouse monoclonal) | Invitrogen | RRID:AB_2609420 | IF (8.3 µg/ml) WM (8.3 µg/ml) |
| Antibody | Anti-perlecan, clone A7L6 (rat monoclonal) | Millipore | RRID:AB_10615958 | IF (1:250) WM (1:250) |
| Antibody | Anti-laminin α5 chain, clone 4C7 (mouse monoclonal) | Abcam | RRID:AB_443652 | IF (1:100) |
| Antibody | Anti-laminin β1 chain, clone 4E10 (mouse monoclonal) | Millipore | RRID:AB_571039 | IF (1:100) |
| Antibody | Anti-laminin S/laminin β2 chain, clone CL2979 (mouse monoclonal) | Novus Biologicals | Cat# NBP-42387 | IF (1:50) WB (1:1000) |
| Antibody | Anti-podocin (rabbit polyclonal) | Millipore | RRID:AB_261982 | IF (1:200) |
| Antibody | Anti-NPHS2 (rabbit polyclonal) | Abcam | RRID:AB_882097 | IF (1:200) |
| Antibody | Anti-integrin β1 chain, clone 9EG7 (rat monoclonal) | **Lenter et al., 1993** | - | IF (1:100) |
| Antibody | Anti-rat IgG conjugated with Alexa Fluor 488 (donkey) | Invitrogen Antibodies | RRID:AB_141709 | IF (1:400) |
| Antibody | Anti-rat IgG conjugated with Alexa Fluor 594 (donkey) | Invitrogen Antibodies | RRID:AB_2535795 | IF (1:400) |
| Antibody | Anti-mouse IgG conjugated with Alexa Fluor 488 (donkey) | Invitrogen Antibodies | RRID:AB_141607 | IF (1:400) |
| Antibody | Anti-mouse IgG conjugated with Alexa Fluor 594 (donkey) | Invitrogen Antibodies | RRID:AB_141633 | IF (1:400) |
| Antibody | Anti-rabbit IgG conjugated with Alexa Fluor 488 (donkey) | Invitrogen Antibodies | RRID:AB_2535792 | IF (1:400) |
| Antibody | Anti-rabbit IgG conjugated with Alexa Fluor 647 (donkey) | Invitrogen Antibodies | RRID:AB_2536183 | IF (1:400) |
| Antibody | Anti-rabbit IgG conjugated with Alexa Fluor 594 (goat) | Invitrogen Antibodies | RRID:AB_141359 | IF (1:400) |
| Antibody | Anti-sheep IgG conjugated with Alexa Fluor 594 | Invitrogen Antibodies | RRID:AB_2534083 | IF (1:400) |
| Antibody | Anti-sheep IgG conjugated with Alexa Fluor 680 | Invitrogen Antibodies | RRID:AB_1500713 | IF (1:400) |
| Antibody | Anti-mouse IgG DyLight 800 4X PEG conjugated (goat) | Cell Signaling Technology | RRID:AB_10697505 | WB (1:1000) |
| Antibody | Anti-rabbit IgG labeled with 10 nm gold (goat) | Abcam | RRID:AB_954434 | Immunogold (1:400) |
| Chemical compound, drug | CHIR99021 | Tocris Bioscience | 4423/10 | - |
| Chemical compound, drug | FGF-9 | PeproTech | 100-23 | - |
| Chemical compound, drug | Heparin | Sigma-Aldrich | H3393 | - |
| Chemical compound, drug | TeSR-E8 medium | STEMCELL | 05991; 05992 | - |
| Chemical compound, drug | STEMdiff APEL 2 medium | STEMCELL | 05270 | - |
| Software, algorithm | Proteome Discoverer v.2.3.0.523 | Thermo Fisher Scientific | RRID:SCR_014477 | - |

## Human fetal kidney

Formaldehyde-fixed paraffin-embedded (FFPE) human fetal kidney sections were provided by the Joint MRC/Wellcome Trust Human Developmental Biology Resource (HDBR; http://hdbr.org). The HDBR

obtains written consent from the donors and has ethics approval (REC reference: 08/H0712/34+5) to collect, store, and distribute human material sampled between 4 and 21 wpc. All experimental protocols were approved by the Institute's Ethical Committee (reference 010/H0713/6) and performed in accordance with institutional ethical and regulatory guidelines.

## Induced pluripotent stem cells

Human iPSCs derived from peripheral blood mononuclear cells (PBMCs) of a healthy individual were generated as previously described (*Wood et al., 2020*). Whole-blood PBMCs were isolated using a Ficoll-Paque (GE17-1440, GE Healthcare) and grown in StemSpan Erythroid Expansion Medium (02692, STEMCELL Technologies) for 8 days before being transduced using CytoTune-iPS 2.0 Sendai virus (A16517, Invitrogen) and grown on vitronectin (A14700, Gibco)-coated plates in ReproTeSR medium (05926, STEMCELL Technologies). When large enough, colonies were manually isolated and grown in TeSR-E8 medium (05991, STEMCELL Technologies). iPSCs derived from patients with AS (see 'Clinical presentation') were generated at the Wellcome Sanger Institute, in collaboration with the Human Induced Pluripotent Stem Cell Initiative (HipSci, https://www.hipsci.org/; *Kilpinen et al., 2017*). Following ethical approval (REC reference 11/H1003/3) and patient consent, dermal fibroblasts were obtained from skin biopsies and programmed to iPSC. For this study, we investigated three members of the same family: a male index patient and his mother carrying a likely pathogenic *COL4A5* variant (c.3695G>A; p.Gly1232Asp, PM1_strong, PM2_moderate, PM5_moderate, PP3_moderate, posterior probability 0.988) and a *COL4A4* variant of uncertain significance (c.3286C>T; p.Pro1096Ser, PM2_moderate, PP3_moderate, posterior probability 0.5), and his sister carrying only the *COL4A5* variant. All human cell lines have been authenticated using short tandem repeat profiling. All experiments were performed with mycoplasma-free cells.

## Clinical presentation

The male index patient, aged 4, presented with recurrent episodes of macroscopic hematuria and persistent microscopic hematuria, and his mother also exhibited microscopic hematuria. He underwent a kidney biopsy, which had normal appearances by light microscopy, and immunohistochemical analysis did not show glomerular deposition of immunoreactants. By electron microscopy, the GBM was thinned in some glomerular capillary loops. In one or two others, the GBM was irregularly thickened with lamination. There were also electron-lucent lacunae between the laminations, some of which contained electron-dense regions, and the ultrastructural changes, in combination with the highly suggestive presentation and family history, confirmed a diagnosis of AS. The patient subsequently developed hypertension and was treated with antihypertensive medications, including renin-angiotensin system blockade, and exhibited slow decline in his kidney function. He eventually progressed to end-stage kidney disease and received a preemptive kidney transplant from his healthy father at the age of 19. Subsequent genetic testing identified a likely pathogenic missense variant in *COL4A5* (c.3695G>A; p.Gly1232Asp) and a variant of unknown significance in *COL4A4* (c.3286C>T; p.Pro1096Ser). Further genetic testing in the family confirmed that both the patient's mother and sister carried one or both variants. His mother had both the *COL4A5* and *COL4A4* variants and exhibited persistent microscopic hematuria and urine protein:creatinine ratios (uPCRs) ranging from 17 to 30 mg/mmol and estimated glomerular filtration rate (eGFR) steady between 74 and 85 ml/min/1.73 m² body surface area over 5 years to most recent follow-up aged 55. The index patient's sister shared only the *COL4A5* variant and exhibited persistent hematuria with proteinuria (uPCR 264 mg/mmol falling to 45–89 mg/mmol on institution of renin-angiotensin system blockade aged 25). Serum creatinine was normal with eGFR > 90 ml/min/1.73 m² body surface area at last follow-up aged 30. The *COL4A4* variant was also detected in the index case's healthy maternal grandmother (in whom urinalysis was normal), but the *COL4A5* variant was absent from both maternal grandparents, suggesting a *de novo* variant in his mother. Pathogenicity was assessed following ACMG coding criteria (*COL4A5* variant: PM1_STR, PM2_MOD, PM5_MOD, PP3_MOD; *COL4A4* variant: PM2_MOD, PP3_MOD).

## Kidney organoid differentiation

We differentiated human iPSCs (*Supplementary file 1*) to 3D kidney organoids as previously described (*Takasato et al., 2015*). iPSCs were maintained at 37°C in TeSR-E8 medium with 25× Supplement (05991, 05992, STEMCELL Technologies) in six-well plates (3516, Corning) coated with vitronectin

(A14700, Gibco). Prior to differentiation (day 0), cells were dissociated with TrypLE (12563029, Thermo Fisher Scientific), counted with a hemocytometer and seeded in vitronectin-coated 24-well plates (3524, Corning) at a density of 35,000 cells/cm$^2$ in TeSR-E8 medium with RevitaCell 10 µl/ ml (A2644501, Gibco). Intermediate mesoderm induction was performed by changing medium after 24 hr to STEMdiff APEL 2 (05270, STEMCELL) with 3% protein-free hybridoma medium (12040077, Gibco) and 8 µM CHIR99021 (4423/10, Tocris Bioscience) for 4 days. On day 5, CHIR99021 was replaced by 200 ng/ml FGF-9 (100-23, PeproTech) and 1 µg/ml heparin (H3393, Sigma-Aldrich). On day 7, cells were dissociated with TrypLE, counted and pelletized into organoids (250,000 cells each) by centrifuging them at 400 × $g$/min four times. Organoids were carefully placed on 0.4 µm Millicell Cell Culture Insert in six-well plates (PICM0RG50, Millipore) and cultured for 1 hr in APEL 2 medium with 5 µM CHIR99021 and subsequently in APEL 2 medium supplemented with 200 ng/ml FGF9 and 1 µg/ml heparin until day 11. From day 12, organoids were grown in STEMdiff APEL 2 without growth factors, with medium changed every 2 days.

## Mice

All mouse handling and experimental procedures were approved by the Animal Ethics Committee of the Institute of Biomedical Sciences (ICB), University of São Paulo (USP), Brazil (reference 019/2015). Two-month-old *Swiss* female mice were housed in an experimental animal facility (ICB, USP), and kept in groups of 3–4 subjects per cage (41 × 34 × 16 cm) at 12 hr light/dark cycle at 25°C, with free access to water and chow. Mating occurred overnight, and females were checked for vaginal plugs on the next morning to determine if mating had occurred and gestation was timed accordingly (E1). Pregnant dams were separated and kept in individual cages (30 × 20 × 13 cm) under the same conditions mentioned previously. Fetuses were collected on E19, following C-section surgery in the pregnant mice under anesthesia with 25 mg/kg avertin (T48402, Sigma-Aldrich). Fetal kidneys were dissected and processed for histological analysis or snap-frozen in liquid nitrogen for proteomic analysis.

## Whole-mount immunofluorescence

Whole organoids were fixed with 2% (wt/vol) paraformaldehyde at 4°C for 20 min, washed with phosphate buffered saline (PBS; D8537, Sigma-Aldrich) and blocked with 1× casein blocking buffer (B6429, Sigma-Aldrich) for 2 hr at room temperature. Samples were incubated at 4°C overnight with primary antibodies diluted in blocking buffer. After thoroughly washing with 0.3% (vol/vol) Triton X-100 in PBS, the samples were incubated at 4°C overnight with Alexa Fluor-conjugated secondary antibodies. Nuclei were stained with Hoechst 33342 solution (B2261, Sigma-Aldrich). Samples were mounted in glass-bottomed dishes (P35G-1.5-10C, MatTek) with ProLong Gold Antifade mountant (P36934, Invitrogen) and imaged with a Leica TCS SP8 AOBS inverted confocal microscope using hybrid detectors with the following detection mirror settings: FITC 494–530 nm; Texas red 602–665 nm; Cy5 640–690 nm. When it was not possible to eliminate fluorescence cross-talk, the images were collected sequentially. When acquiring 3D optical stacks, the confocal software was used to determine the optimal number of Z sections. Only the maximum intensity projections of these 3D stacks are shown in the results. 3D image stacks were analyzed with ImageJ v 1.53g software (Rasband, W.S., ImageJ, U.S. National Institutes of Health, Bethesda, MD, USA; available at https://imagej.nih.gov/ij/, 1997–2018).

## Histology and immunofluorescence

For light microscopy, FFPE sections were stained with hematoxylin and eosin (H&E) for morphological analysis. Images were acquired on a 3D-Histech Pannoramic-250 microscope slide scanner (Zeiss) using the Case Viewer software (3D-Histech). For immunofluorescence microscopy, either cryosections or FFPE were subjected to nonspecific binding site blocking with 10% normal donkey serum in 1% BSA/PBS, and treated with primary and secondary antibody solutions (see Key resources table). FFPE samples were submitted to heat-induced antigen retrieval with 10 mM sodium citrate buffer (pH 6.0) in a microwave for 15 min and treated with 0.1 M glycine/6 M urea solution for 30 min at room temperature prior to blocking. The slides were mounted and analyzed with a Zeiss Axioimager.D2 upright microscope equipped with a Coolsnap HQ2 camera (Photometrics). Images were acquired with the Micromanager Software v1.4.23 and processed using ImageJ.

## Electron microscopy

Whole-mount primary antibody labeling was performed as described above. After overnight incubation at 4°C with a rabbit pAb anti-laminin antibody (ab11575, Abcam), organoids were washed with PBS-Triton and incubated overnight at 4°C with a goat anti-rabbit IgG labeled with 10 nm gold (ab39601, Abcam) diluted 1:400. Samples were then fixed with 4% paraformaldehyde and 2.5% (wt/vol) glutaraldehyde (Agar Scientific, UK) in 0.1 M HEPES (H0887, Sigma-Aldrich) pH 7.2, and postfixed with 1% osmium tetroxide (R1024, Agar Scientific) and 1.5% potassium ferrocyanide (214022, The British Drug House, Laboratory Chemicals Division) in 0.1 M cacodylate buffer (R1102, Agar Scientific) pH 7.2 for 1 hr, then with 1% uranyl acetate (R1100A, Agar Scientific) in water for overnight. Samples were dehydrated, embedded with low-viscosity medium-grade resin (T262, TAAB Laboratories Equipment Ltd) and polymerized for 24 hr at 60°C. For transmission electron microscopy, sections were cut with a Reichert Ultracut ultramicrotome and observed with a FEI Tecnai 12 Biotwin microscope at 80 kV accelerating voltage equipped with a Gatan Orius SC1000 CCD camera.

## SDS-PAGE and immunoblotting

Organoid samples were homogenized in ice-cold Pierce IP Lysis Buffer Proteins (87787, Thermo Fisher) supplemented with EDTA-free protease inhibitor cocktail (04-693-159-001, Roche) to extract proteins. Then, proteins were resolved by SDS-PAGE in a NuPAGE 4–12% Bis-Tris gel (NP0322, Invitrogen) and wet-transferred to a nitrocellulose membrane (Z612391, Whatman). Gel loading was assessed by Ponceau S staining (P7170, Sigma-Aldrich). Membranes were blocked with Odyssey blocking buffer (927-40000, LI-COR) for 1 hr and probed with specific primary and secondary antibodies (see Key resources table) diluted in Tris-buffered saline solution (TBS). Protein bands were visualized using the Odyssey CLx Imaging System (LI-COR Biosciences), and background-corrected band optical densitometry was determined using ImageJ.

## Sample enrichment for proteomics analysis

Kidney organoids samples (days 14, 18, and 25 of differentiation) were pooled accordingly (n = 3 pools per time point), and E19 mouse fetal kidneys (n = 6) were enriched for matrix proteins as previously described (*Lennon et al., 2014*). Briefly, samples were manually homogenized and incubated in a Tris-buffer (10 mM Tris pH 8.0, 150 mM NaCl, 25 mM EDTA, 1% Triton X-100, and EDTA-free protease inhibitor cocktail) for 1 hr to extract soluble proteins. The supernatant (fraction 1) was collected by centrifugation (at 14,000 × $g$ for 10 min), and the remaining pellet was resuspended in an alkaline detergent buffer (20 mM $Na_4OH$, 0.5% in PBS-Triton) and incubated for 1 hr to solubilize and disrupt cell-matrix interactions. The supernatant (fraction 2) was collected by centrifugation and the pellet treated with 0.4 µg Benzonase (E1014-25KU, Sigma-Aldrich) in PBS (D8537, Sigma-Aldrich) for 30 min at room temperature to remove DNA/RNA contaminants. After inactivating Benzonase at 65°C for 20 min, samples were centrifuged and the remaining pellet was resuspended in 5× reducing sample buffer (100 mM Tris pH 6.8, 25% glycerol, 10% SDS, 10% β-mercaptoethanol, 0.1% bromophenol blue) to yield the ECM fraction. Fractions 1 and 2 were combined (1:1) into a cellular fraction.

## Laser microdissection microscopy

E19 mouse kidneys (n = 4) were embedded in OCT for cryosectioning. 10-µm-thick cryosections were acquired and placed onto MMI membrane slides (50102, Molecular Machines and Industries), fixed with 70% ethanol, and stained with H&E to allow visualization of maturing glomeruli using a ×40/0.5 FL N objective. 150 glomeruli (per samples) were laser-microdissected around the Bowman's capsule using an Olympus IX83 Inverted Fluorescence Snapshot Microscopy equipped with MMI CellCut Microdissection system and the MMI CellTools software v.5.0 (Molecular Machines and Industries). Laser settings were speed = 25 µm/s, focus = 16.45 µm, and power = 72.5%. Sections were collected onto sticky 0.5 ml microtube caps (Molecular Machines and Industries) and stored at –80°C.

## Trypsin digestion

For in-gel digestion, protein samples from E19 mouse kidneys were briefly subjected to SDS-PAGE to concentrate proteins in the gel top and stained with Expedeon InstantBlue (Z2, Fisher Scientific). After distaining, gel-top protein bands were sliced and transferred to a V-bottomed perforated 96-wells plate (Proxeon) and incubated with 50% acetonitrile (ACN) in digestion buffer (25 mM $NaHCO_3$) for

30 min at room temperature. After centrifuging (1500 rpm for 2 min), gel pieces were shrunk with 50% ACN and completely dried by vacuum centrifugation for 20 min at –120°C. After rehydrating gel pieces, proteins were reduced with 10 mM dithiothreitol (DTT; D5545, Sigma-Aldrich) in digestion buffer for 1 hr at 56°C and alkylated with 55 mM iodoacetamide (IA; I149, Sigma-Aldrich) in digestion buffer for 45 min at room temperature in the dark and spun down. Gel pieces were shrunk with 100% ACN and dried by vacuum centrifugation. Protein digestion with 1.25 ng/l trypsin (V5111, Promega) was carried out at 37°C overnight, followed by centrifugation to collect the resulting peptides. Finally, samples were dried by vacuum centrifugation and resuspended in 50% ACN in 0.1% formic acid. For in-solution digestion, samples were sonicated in lysis buffer (5% SDS in 50 mM TEAB pH 7.5) using a Covaris LE220+ Focused Ultrasonicator (Covaris), reduced with 5 mM DTT for 10 min at 60°C, and alkylated with 15 mM IA for 10 min at room temperature in the dark. After quenching residual alkylation reaction with 5 mM DTT, samples were spun down, acidified with 1.2% formic acid, and transferred to S-Trap Micro Spin columns (Protifi). Contaminants were removed by centrifugation, and protein digestion with 0.12 g/l trypsin (in 50 mM TEAB buffer) was carried out 1 hr at 47°C. Trapped peptides were thoroughly washed with 50 mM TEAB, spun down, washed with 0.1% formic acid, and eluted from the S-trap columns with 30% ACN in 0.1% formic acid solution.

## Offline peptide desalting

Peptide samples were incubated with 5.0 mg Oligo R3 reverse-phase beads (1133903, Applied Biosystems) in 50% ACN in a 96-well plate equipped with 0.2 m polyvinylidene fluoride membrane filter (3504, Corning). After centrifuging, the bead-bound peptides were washed twice with 0.1% formic acid, spun down, and eluted with 30% ACN in 0.1% formic acid. Retrieved peptides were dried by vacuum centrifugation and sent to the Bio-MS Core Research Facility (Faculty of Biology, Medicine and Health, University of Manchester) for MS analysis.

## MS data acquisition and analysis

Peptide samples were analyzed by liquid chromatography (LC)-tandem mass spectrometry using an UltiMate 3000 Rapid Separation LC (RSLC, Dionex Corporation, Sunnyvale, CA) coupled to a Q Exactive Hybrid Quadrupole-Orbitrap (Thermo Fisher Scientific, Waltham, MA) mass spectrometer. Peptides were separated on a CSH C18 analytical column (Waters) using a gradient from 93% A (0.1% formic acid in water) and 7% B (0.1% formic acid in ACN) to 18% B over 57 min followed by a second gradient to 27% B over 14 min both at 300 nl/min. Peptides were selected for fragmentation automatically by data-dependent acquisition. Raw spectra data were acquired and later analyzed using Proteome Discoverer software v.2.3.0.523 (Thermo Fisher Scientific). MS data were searched against the SwissProt and TrEMBL databases (v. 2018_01; OS = *Mus musculus* for mouse samples; OS = *Homo sapiens* for kidney organoids) using SEQUEST HT and Mascot (https://www.matrixscience.com/) search tools. Tryptic peptides with ≤1 missed cleavage were considered for the search, and mass tolerance for precursor and fragment ions were 10 ppm and 0.02 Da, respectively. Carbamidomethylation of cysteine was as fixed modification, oxidation of methionine, proline, and lysine, and N-terminal acetylation as dynamic modifications. False discovery rate (FDR) for peptide/protein identifications was set to 1%, and protein validation was performed using Target/Decoy strategy. Label-free protein abundances were determined based on precursor ion intensity and relative changes in protein abundance by calculating abundance ratios accordingly. Results were filtered for significant FDR master proteins identified with ≥1 unique peptide detected in 2/3 of replicates. The MS proteomics data have been deposited to the ProteomeXchange Consortium via the PRIDE partner repository (*Perez-Riverol et al., 2019*) with the dataset identifiers: PXD025838, PXD025874, PXD025911 and PXD026002.

## Enrichment and interactome analyses

Gene Ontology (GO) enrichment analysis was performed using the DAVID bioinformatics resource v.6.8 (*Huang et al., 2009*; https://david.ncifcrf.gov), and term enrichment was determined through Fisher's exact test with Benjamini–Hochberg correction, with a term selected as enriched when FDR < 0.1. Pathway enrichment was performed for proteins differentially expressed using the Reactome database (*Jassal et al., 2020*; https://reactome.org/). To generate interactome figures, a list of proteins was uploaded to STRING v.11.0 (*Szklarczyk et al., 2015*) to obtain a collection of high-confident

reported protein-protein interactions (combined score ≥70%), which was further uploaded into Cytoscape v.3.8.1 (*Shannon et al., 2003*) to customize the interactomes.

## Single-cell RNA-sequencing analysis

We selected three published single-cell RNA-sequencing datasets generated from kidney organoids (GSE114802), fetal and adult human kidneys (EGAS00001002325, EGAS00001002553), and mouse fetal kidney (GSE108291) to identify the cellular origins of BM genes. We first removed the low-quality cells from the dataset to ensure that the technical noise does not affect the downstream analysis. We also removed the lowly expressed genes as they do not give much information and are unreliable for downstream statistical analysis (*Bourgon et al., 2010*). In order to account for the various sequencing depth of each cell, we normalized the raw counts using the deconvolution-based method (*Lun et al., 2016a*). We then identified the genes that had high variance in their biological component and used these genes for all downstream analysis. We then applied PCA and took the first 14 components of PCA as input to tSNE and used the first 2 components of tSNE to visualize our cells. The cells were then grouped into their putative clusters using the dynamic tree cut method. We used the *findMarkers* function from *Scran* package to identify the marker genes for each of the clusters (*Lun et al., 2016b*). *findMarkers* uses *t*-test for the statistical test and reports *p*-value of the high-rank genes that are differentially expressed between the group and all other groups. These marker genes were then used to manually annotate the cell types of a cluster (see *Supplementary file 3* for clustering annotation details). We also applied SingleR to define the cell types based on matched with annotated bulk datasets (*Aran et al., 2019*). We used the *plotDots* function from scater package to produce the dot plots (*McCarthy et al., 2017*).

## Statistical analysis

Statistical analysis was carried out within Proteome Discoverer using an in-built two-way ANOVA test with post-hoc Benjamini–Hochberg correction. PCA and unsupervised hierarchical clustering based on a Euclidean distance-based complete-linkage matrix were performed using RStudio v. 1.2.5042 (http://rstudio.com) with the ggplot2 package v.3.3.2 (https://ggplot2.tidyverse.org) that was also used to generate PCA plots and heat maps. For the integrated proteomic analysis, previously published human glomerular and kidney tubulointerstitial data (PRIDE accession PXD022219) was reprocessed with Proteome Discoverer to allow direct comparisons with newly acquired proteomic data in this study. Then, kidney organoid, mouse, and human proteomics datasets were compared using Spearman's rank correlation. Dataset comparisons, for both cellular and ECM cellular fractions, were performed separately for the matrisome proteins only and BM proteins only. The Complex-Heatmap package v2.2.0 (*Gu et al., 2016*; http://bioconductor.org/packages/release/bioc/html/ComplexHeatmap.html) was used to generate correlation plots.

## Acknowledgements

We acknowledge Faris Tengku, who helped with the generation of wild-type iPSC, Joseph Luckman, who helped to optimize immunofluorescence protocols, Karen Leigh Price and Maria Kolatsi-Joannou, who helped with the human histology, and Anna-Li, who helped with the description of the genetic variants using the ACMG criteria, and staff from the Biomolecular Analysis, Biological Mass Spectrometry, Bioimaging, and Electron Microscopy Facilities (University of Manchester) for advice and assistance. The mass spectrometer and microscopes used in this study were purchased with grants from the Biotechnology and Biological Sciences Research Council, Wellcome Trust, and the University of Manchester Strategic Fund. Mass spectrometry was performed at the Biomolecular Analysis Core Facility (Faculty of Life Sciences, University of Manchester), and we thank David Knight, Ronan O'cualain, and Stacey Warwood for advice and technical support, and Julian Selley for bioinformatic support. The iPSC lines were generated at the Wellcome Trust Sanger Institute under the Human Induced Pluripotent Stem Cell Initiative (HipSci) funded by a grant (WT098503) from the Wellcome Trust and Medical Research Council.

## Additional information

### Funding

| Funder | Grant reference number | Author |
| --- | --- | --- |
| Wellcome Trust | 202860/Z/16/Z | Rachel Lennon |
| Kidney Research UK | RP52/2014 | Pinyuan Tian<br>Rachel Lennon |
| São Paulo Research Foundation | 2015/02535-2 | Mychel RPT Morais |
| Global Challenges Research Fund | | Mychel RPT Morais |
| Veterans Affairs | 1I01BX002196-01 | Roy Zent |
| NIHR Biomedical Research Centre | | David A Long |
| Wellcome Trust | 220895/Z/20/Z | David A Long |
| Medical Research Council | MR/P018629/1 | David A Long |
| Veterans Affairs | DK069221 | Mychel RPT Morais |
| Medical Research Council | MR/J003638/1 | David A Long |
| São Paulo Research Foundation | 2017/26785-5 | Mychel RPT Morais |

The funders had no role in study design, data collection and interpretation, or the decision to submit the work for publication.

### Author contributions

Mychel RPT Morais, Conceptualization, Data curation, Formal analysis, Methodology, Provided mouse samples, Resources, Visualization, Writing - original draft, Writing - review and editing; Pinyuan Tian, Conceptualization, Data curation, Formal analysis, Methodology, Resources, Writing - original draft, Writing - review and editing; Craig Lawless, Syed Murtuza-Baker, Data curation, Formal analysis, Single cell RNA sequencing analyses, Single cell RNA sequencing analyses, Writing - review and editing; Louise Hopkinson, Investigation, Writing - original draft, Writing - review and editing; Steven Woods, Aleksandr Mironov, Data curation, Formal analysis, Writing - review and editing; David A Long, Provided fetal human kidney sections, Resources, Writing - review and editing; Daniel P Gale, Provided patient samples for iPSC generation., Resources, Writing - original draft, Writing - review and editing; Telma MT Zorn, Provided fetal mouse kidney samples, Resources; Susan J Kimber, Resources, Supervision, Writing - review and editing; Roy Zent, Conceptualization, Supervision, Writing - review and editing; Rachel Lennon, Conceptualization, Funding acquisition, Methodology, Writing - original draft, Writing - review and editing

### Author ORCIDs

Mychel RPT Morais (iD) http://orcid.org/0000-0001-5237-9524
Pinyuan Tian (iD) http://orcid.org/0000-0001-6080-5378
Louise Hopkinson (iD) http://orcid.org/0000-0003-1758-4201
David A Long (iD) http://orcid.org/0000-0001-6580-3435
Daniel P Gale (iD) http://orcid.org/0000-0002-9170-1579
Rachel Lennon (iD) http://orcid.org/0000-0001-6400-0227

### Ethics

Human fetal kidney sections were provided by the Joint MRC/Wellcome Trust Human Developmental Biology Resource (HDBR) (http://hdbr.org). The HDBR obtains written consent from the donors and has ethics approval (REC reference: 08/H0712/34+5) to collect, store and distribute human material sampled between 4 and 21 weeks post conception. All experimental protocols were approved by the Institute's Ethical Committee (reference 010/H0713/6) and were performed in accordance with institutional ethical and regulatory guidelines.

All mouse handling and experimental procedures were approved by the Animal Ethics Committee of the Institute of Biomedical Sciences (University of São Paulo, Brazil; reference 019/2015). This was performed in accordance with recommendations from the current Brazilian legislation. All surgery was performed under avertin anesthesia.

### Decision letter and Author response
Decision letter https://doi.org/10.7554/eLife.73486.sa1
Author response https://doi.org/10.7554/eLife.73486.sa2

---

## Additional files

### Supplementary files
• Supplementary file 1. Human fetal kidney and human induced pluripotent stem cell (hiPSC) general information.

• Supplementary file 2. Human kidney organoid proteome and matrix proteins.

• Supplementary file 3. Single-cell RNA-sequencing kidney datasets: cell clustering and expression data.

• Supplementary file 4. Embryonic day 19 mouse maturing glomerulus proteome and matrix proteins.

• Supplementary file 5. Embryonic day 19 mouse kidney proteome and matrix proteins.

• Supplementary file 6. Human adult kidney glomerular and tubulointerstitial proteome and matrix proteins.

• Transparent reporting form

### Data availability
The mass spectrometry proteomics data have been deposited to the ProteomeXchange Consortium via the PRIDE partner repository (Perez-Riverol et al., 2019) with the dataset identifiers: PXD025838, PXD025874, PXD025911 and PXD026002. This project also contains the following source data hosted at: https://doi.org/10.6084/m9.figshare.c.5429628https://doi.org/10.6084/m9.figshare.c.5429628 Figure 1 Original IF Images: B Whole-mount immunofluorescence for kidney cell types; F Representative whole mount immunofluorescence images of wild-type and Alport kidney organoids; G Immunofluorescence for LAMB2. Figure 1 Original light microscope Images: C Representative photomicrographs of day 18 kidney organoids (left) and human and mouse fetal kidneys (right). Figure 1 Original TEM Images: D Transmission electron micrographs of tubular BM in day 25 kidney organoid and E19 mouse fetal kidney. Figure 1 Original western blotting image: H Immunoblotting for LAMB2 using total lysates from wild-type and Alport organoids. Figure 2 Original IF Images: A Confocal immunofluorescence microscopy of wild-type kidney organoids; B perlecan and nidogen on days 11, 18 and 25 of differentiation. Figure 4 Original IF Images: A Immunofluorescence for key type IV collagen and laminin isoforms showing their emergence and distribution in kidney organoid BM; D Immunofluorescence for specific collagen IV isoforms in maturing glomeruli in E19 mouse kidney and in glomerular structures (indicated by dashed lines) in day 25 organoids. Figure 1-figure supplement 2A Original TEM photomicrographs: A Transmission electron microscopy of day 25 kidney organoids shows advanced differentiation of glomerular structures. Figure 1 - figure supplement 1. Morphological characteristics of wild-type kidney organoids, fetal human kidney, and Alport kidney organoids. Figure 1-figure supplement 2B Original TEM photomicrographs: B Transmission electron microscopy of day 25 kidney organoids shows advanced differentiation of glomerular structures. Figure 1-figure supplement 1C Original IF images: C Immunofluorescence for integrin beta-1 (ITGB1) in day 25 kidney organoid (wild-type). Anti-panlaminin or anti-collagen IV antibodies were used to label basement membranes. Figure 3 - figure supplement 1. Time course proteomic analysis of kidney organoid differentiation. Figure 4 - figure supplement 1. Single cell-RNA sequencing data analysis of human kidney organoids. Figure 5 - figure supplement 1. Proteomic analysis of E19 mouse fetal kidney and correlational comparison with kidney organoid proteomics. Figure 6 - figure supplement 1. Integrated correlational analysis of organoid and in vivo kidney datasets. Supplementary file 1 Human fetal kidney and hiPSC general information. Supplementary file 2 Human kidney organoid proteome and matrix proteins.Supplementary file 3 scRNA-seq kidney datasets- cell clustering and expression

data. Supplementary file 4 E19 mouse maturing glomerulus proteome and matrix proteins. Supplementary file 5 E19 mouse kidney proteome and matrix proteins. Supplementary file 6 Human adult kidney glomerular and tubulointerstitial proteome and matrix proteins.

The following datasets were generated:

| Author(s) | Year | Dataset title | Dataset URL | Database and Identifier |
|---|---|---|---|---|
| Tian P, Lawless C, Baker SM, Hopkinson L, Woods S, Mironov A, Long DA, Gale D, Kimber S, Zent R, Lennon R, Morais MRT, Zorn TMT | 2021 | Proteomic analysis of induced pluripotent stem cell - derived kidney organoids | https://www.ebi.ac.uk/pride/PXD025838 | PRIDE, PXD025838 |
| Tian P, Lawless C, Baker SM, Hopkinson L, Woods S, Mironov A, Long DA, Gale D, Kimber S, Zent R, Lennon R, Morais MRT, Zorn TMT | 2021 | Proteomic analysis of healthy and hyperglycemic murine fetal kidneys | https://www.ebi.ac.uk/pride/PXD025874 | PRIDE, PXD025874 |
| Tian P, Lawless C, Baker SM, Hopkinson L, Woods S, Mironov A, Long DA, Gale D, Kimber S, Zent R, Lennon R, Morais MRT, Zorn TMT | 2021 | Proteomic analysis of laser - microdissected murine fetal glomeruli | https://www.ebi.ac.uk/pride/PXD025911 | PRIDE, PXD025911 |
| Tian P, Lawless C, Baker SM, Hopkinson L, Woods S, Mironov A, Long DA, Gale D, Kimber S, Zent R, Lennon R, Morais MRT, Zorn TMT | 2021 | Human kidney proteomics, glomerular and tubular cellular and matrix fractions | https://www.ebi.ac.uk/pride/PXD026002 | PRIDE, PXD026002 |
| Tian P, Lawless C, Murtaza-Baker S, Hopkinson L, Woods S, Mironov A, Long A, Gale A, Zorn TMT, Zent R, Lennon R, Morais MRT | 2021 | Kidney organoids: A system to study human basement membrane assembly in health and disease | https://doi.org/10.6084/m9.figshare.c.5429628.v3 | figshare, 10.6084/m9.figshare.c.5429628.v3 |

The following previously published datasets were used:

| Author(s) | Year | Dataset title | Dataset URL | Database and Identifier |
|---|---|---|---|---|
| Young MD, Mitchell TJ, Braga FAV, Tran MGP, Stewart BJ, Ferdinand JR, Collord G, Botting RA, Popescu D, Loudon KW, Vento-Tormo R, Stephenson E, Cagan A, Farndon S, Castillo MD, Herrera V, Guzzo C, Richoz N, Mamanova L, Aho T, Armitage JN, Riddick ACP, Mushtaq I, Farrell S, Rampling D, Nicholson J, Filby A, Burge J, Lisgo S, Maxwell PH, Lindsay S, Warren AY, Stewart GD, Sebire N, Coleman N, Haniffa M, Teichmann SA, Clatworthyand M, Behjati S | 2018 | Kidney single cell study | https://ega-archive.org/studies/EGAS00001002325 | EGA, EGAS00001002325 |
| Combes AN, Zappia L, Er PX, Oshlack A, Little MH | 2019 | Single cell RNA-Seq of four human kidney organoids | https://www.ncbi.nlm.nih.gov/geo/query/acc.cgi?acc=GSE114802 | NCBI Gene Expression Omnibus, GSE114802 |
| Young MD, Mitchell TJ, Braga FAV, Tran MGP, Stewart BJ, Ferdinand JR, Collord G, Botting RA, Popescu D, Loudon KW, Vento-Tormo R, Stephenson E, Cagan A, Farndon S, Castillo MD, Herrera V, Guzzo C, Richoz N, Mamanova L, Aho T, Armitage JN, Riddick ACP, Mushtaq I, Farrell S, Rampling D, Nicholson J, Filby A, Burge J, Lisgo S, Maxwell PH, Lindsay S, Warren AY, Stewart GD, Sebire N, Coleman N, Haniffa M, Teichmann SA, Clatworthyand M, Behjati S | 2018 | Pilot Fetal Cell Atlas RNAseq | https://ega-archive.org/datasets/EGAD00001004305 | EGA, EGAS00001002553 |

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
