## [Editor Report]

Kidney organoid cultures derived from human induced pluripotent stem cells represent a new tool with which to study renal morphogenesis in both normal and pathological states. In the current study, the authors have combined morphological evaluation with proteomics to elucidate aspects of the temporal sequence of basement membrane composition during normal renal development and in the setting of a pathogenic collagen type IV α5 chain variant associated with Alport syndrome, an inherited kidney disease. This model system may help us to better understand the normal processes of basement membrane development and the pathogenesis of inherited diseases that affect renal basement membrane composition.

---

## [Decision Letter]

**Decision letter after peer review:**

Thank you for submitting your article "Kidney organoids: A system to study human basement membrane assembly in health and disease" for consideration by *eLife*. Your article has been reviewed by 3 peer reviewers, and the evaluation has been overseen by a Reviewing Editor and Martin Pollak as the Senior Editor. The following individual involved in review of your submission has agreed to reveal their identity: Peter Yurchenco (Reviewer #3).

Essential revisions:

1) Please add a figure showing the ultrastructure of day 25 glomeruli by EM.

2) Please add immunostaining data of the late-stage organoids to demonstrate the distribution pattern of integrins and dystroglycan.

3) The discussion needs to be expanded to address the following points:

a) It should more clearly explain what is truly novel here---what is significantly new compared to earlier studies including a 2018 study co-authored by Dr. Lennon.

b) It should review the strengths and weaknesses of the organoid model, including consideration of the issue of an apparent lack of cell-type specificity of Lma5, Lmb2 and Col-IV a3/a4/a5 staining as it appears in the organoids.

c) Please comment on what degree the organoid kidney replicates differentiation in mice and humans.

4) Please add either text and/or a figure that reviews the developmental milestones in kidney morphogenesis, describing/showing how the organoid and in vivo developmental systems compare.

*Reviewer #1 (Recommendations for the authors):*

This manuscript is too long – the Method is far too detailed. Can you please shorten it? For example I don't think you need to include the washing steps or the method for western blots…

We want to know if you were able to demonstrate an abnormal GBM in the Alport organoid? What did the basement membrane look like?

Is the organoid a model only for the capillary network or what else does it comprise?

The Discussion could be more informative. The manuscript does not include a lot of discussion – for example, you do not compare your strategy with what others have done and shown the novelty and the advantage of your method. You also do not consider the limitations of your approach? What were its weaknesses?

We already know a bit about the expression of the different basement membrane component and the switch – can you emphasise what is novel in your work? The expression of the components is not very novel – it is now available on line in KIT? What do your results add to this?

You probably do not need to abbreviate basement membrane to BM. Usually we now refer to collagen IV α3 chain rather than collagen α3(IV) chain.

A diagram of the different stages of kidney development would be useful…with the proteins in control and expressed in each.

Usually we do not refer to people with a disease as a 'case'. This description of the boy could be much shorter. You have not indicated that you had ethics approval for this?

You need to describe pathogenic variants using the ACMG criteria to indicate that they are pathogenic. If P1096S is pathogenic then that complicates matters…

*Reviewer #2 (Recommendations for the authors):*

My key suggestion would be to bring out an emphasis that this is a model that best correlates and is applicable to kidney disease, as kidney BMs will have variation from other BMs. This is not a detraction from this work, I think this creates a new stimulus to investigate BMs across tissues and to create tractable models similar to the kidney organoids presented here.

*Reviewer #3 (Recommendations for the authors):*

The current submissions addresses temporal and tissue-specific BM changes during organoid kidney development. Day 25 kidney organoids contained tubules, stroma, and glomeruli with partial resemblance to (mouse) E19 kidney. Tubular and glomerular BMs are seen to form, the latter showing the expected switch from α-1/β-1 laminins to α-5/β-2 laminins, and alpha1/2 type IV collagens to alpha3-containing type IV collagens required for glomerular maturation.

Specific questions/comments:

1. Figure 2 and text: [a] Figure 2A, Microvascular endothelium: A key feature of the study in the replacement of Lm111 by Lm521 and Col-IV alpha1/2 by Col-IV α 3/4/5 in the developing glomerulus. This occurs in normal maturation as the glomerulus gains a vascular tuft. In this study, a regent specific to CD31 (PECAM-1; found on endothelial cells, but also on macrophages, granulocytes) is used to detect endothelial cells. Figure 2A shows increasing CD31 staining at days 11, 18 and 25. While most of the staining at day 25 appears between or at the edges of tubular-like structures, a single narrow stained-streak that co-localizes with laminin is present within a glomerulus. This may reflect the early appearance of a vascular tuft with a laminin coat. Are similar entities present in other glomeruli? If so, it would help to show this. Further, this reviewer feels it is important to show the ultrastructure of such glomeruli to gain insight into the organization (e.g. adjacent cells forming lumens) and surrounding structures of the C31-positive cells. Is there an early electron-dense BM structure lining these cells and, in particular, Are there both podocyte-derived and endothelial cell derived BMs?

2. If vascular-tuft invasion of day 25 organoid glomeruli is only just beginning to occur (as suggested by the images), does it not follow that this organoid stage corresponds to an earlier mouse stage than that of the E19 mouse fetus? Given mice are born around E19.5 fully capable of forming urine, it is hard to imagine the implied correspondence. Perhaps the day 25 organoids reflect an earlier stage corresponding to the start of vascular invasion of the glomerulus (about E16.5).

3. Figure 1G: An Alport patient iPSC line with an X-linked missense variant of COL4A5 was also evaluated and it was found that LAMB2 expression was increased in extra-glomerular BM. Such laminin compensation can apparently be seen in patients. The laminin beta2 increase appears to occur in all BMs of the organoids, not just the glomeruli. Is this what is seen in human patients?

4. Figure 4. Distribution of BM components: [a] By day 18, laminin beta2 is seen to be present in the BMs of glomeruli and tubules. However, the expectation is that beta2 should largely be confined to glomeruli and blood vessels. BMs do not assemble on all cell surfaces as a generality (e.g. compare the basal side of epithelia with stromal fibroblasts). Further, specific BM components are only incorporated into select BMs. Such specificity can be a consequence of a lack of cognate receptors (especially relevant for laminins), other binding sites, and/or a consequence of a barrier to diffusion of secreted BM components from the source cells. So why is there a difference between mouse kidneys and organoid kidneys?

[b] Laminin α 5 (Lma5) also appears to be widely distributed within the kidney organoid structures. During kidney development, Lma5 is strongly capillary loop/developing GBM localized (and also in blood vessels), although on can see some around tubules as well. In the organoids, the distribution appears quite even for glomerular, tubular and other structures.

[c] One therefore has to wonder whether the same receptors (integrins, dystroglycan, other) are expressed in normal developing kidney as compared to organoid kidneys.

[d] Collagen-IV alpha3 appears to be as widely distributed as collagen IV alpha1. Normally, Col4a345 is strongly localized to the capillary loops and GBM. The mechanism for the more restricted assembly of these subunits is not well understood as far as I am aware. Discussion of this would be helpful.

5. Lines 70-72 of text. The authors state that "there is limited understanding about BM assembly". This reviewer would argue that actually there is a substantial body of published evidence derived from biochemical, cell biological and genetic approaches. This body is relevant to interpret the organoid/proteomic findings. In any case, different approaches address different aspects of BM assembly, extending from elucidation of binding interactions among components and with receptors to site-specific turnover. Renal organoid differentiation can reveal a host of new information; however, it probably is not the best tool to establish which components bind to each other via specific domain interactions, or the molecular mechanisms that underlie the interactions. To this reviewer, the real kidney advance of the study is a new way to understand the dynamic changes that occur during assembly in a temporal and cell-type specific manner.

6. It would be useful to enumerate both what the authors consider the strengths and limitations of the organoid kidney/proteomic system. It would also be useful to more explicitly state the advances achieved since the 2018 publication.

---

## [Author Response]

Essential revisions:1) Please add a figure showing the ultrastructure of day 25 glomeruli by EM.

We conducted additional electron microscopy of Day 25 organoids and we have included the images in a revised Figure 1 (panel 1D), and a dedicated Figure 1—figure supplement 2. These new images show glomerular and tubular structures within the organoids and highlight basement membrane structures adjacent to podocyte primary processes.

2) Please add immunostaining data of the late-stage organoids to demonstrate the distribution pattern of integrins and dystroglycan.

Thank you for this suggestion. We have added new immunofluorescence data to demonstrate the colocalization integrin β 1 and laminin (Figure 1—figure supplement panel 1C). We also acquired and tested a dystroglycan antibody (DAG-6F4 from Developmental Studies Hybridoma Bank, DHSB) but were not able to detect an immunofluorescence signal with two experimental repeats. However, we detected dystroglycan in the organoid proteomic data all time points, increasing from day 14 to day 25 indicating the presence of this laminin receptor in kidney organoids. The localization studies will require further optimization, but we do not think this data is essential for the interpretation of our manuscript.

3) The discussion needs to be expanded to address the following points:a) It should more clearly explain what is truly novel here---what is significantly new compared to earlier studies including a 2018 study co-authored by Dr. Lennon.

We have revised the discussion to highlight the novel findings from this study. The advance from the 2018 study is the analysis of whole kidney organoids during the differentiation time course as opposed to the analysis of isolated organoid glomeruli at a single time point.

b) It should review the strengths and weaknesses of the organoid model, including consideration of the issue of an apparent lack of cell-type specificity of Lma5, Lmb2 and Col-IV a3/a4/a5 staining as it appears in the organoids.

We have reinforced statements on the strengths and weaknesses of the organoid model for studying basement membrane assembly.

c) Please comment on what degree the organoid kidney replicates differentiation in mice and humans.

In our study we focussed on matrix composition and structure in kidney organoids by comparison to mouse and human kidney tissue. There are similarities in both the complex composition of matrix and its organisation into basement membrane structures. To illustrate these similarities, we created a new summary schematic as Figure 7. Overall, we consider kidney organoids as a complementary and adjustable system to investigate the early stages of basement membrane assembly during kidney development.

4) Please add either text and/or a figure that reviews the developmental milestones in kidney morphogenesis, describing/showing how the organoid and in vivo developmental systems compare.

Thank you for this suggestion. We have added a summary Figure 7 to illustrate the developmental milestones in kidney morphogenesis across the organoid, in vivo mouse, and human systems.

Reviewer #1 (Recommendations for the authors):This manuscript is too long – the Method is far too detailed. Can you please shorten it? For example I don't think you need to include the washing steps or the method for western blots…

Thank you for raising this issue. We have carefully revised and shortened the article overall and we have transferred essential detailed technical information to the Supplemental Notes section.

We want to know if you were able to demonstrate an abnormal GBM in the Alport organoid? What did the basement membrane look like?

Thank you for this question. In Figure 1G, we show increased deposition of a LAMB2 within glomerular-like structures in both wild-type and Alport kidney organoids, but no apparent change in basement membrane structure at the level of light microscopy. Since the focus of this first paper is on normal basement membrane assembly, we have not included electron microscopy data on the Alport organoids, and this is the major focus of another project in our laboratory. However, we conducted additional electron microscopy of healthy Day 25 organoids and we have included these images in a revised Figure 1 (panel 1D), and a new dedicated Figure 1—figure supplement 2. We observed basement membrane-like structures, but these are not as established as a typical glomerular basement membrane (GBM) in vivo. The development of a mature GBM is likely to require capillary blood flow. Indeed, we and others have shown further GBM maturation following organoid implantation into immunodeficient mice (PMID: 29429961, PMID: 2950308).

Is the organoid a model only for the capillary network or what else does it comprise?

Through the differentiation protocol kidney organoids pattern into both glomerular and tubular structures with markers of multiple cell types from within these distinct compartments. At least 20 types of kidney cells have been identified in kidney organoids by single-cell transcriptomics and immunofluorescence including endothelial, proximal tubules, connecting segment/ureteric epithelium, and podocytes.

Furthermore, kidney organoids have appropriate nephron morphology including glomeruli, proximal tubules, distal tubules, and loops of Henle, a connecting endothelial network with perivascular cells is also detected. Therefore, the organoid system could be used to investigate matrix assembly in all compartments of the developing kidney. We focussed on the glomerular basement membrane since there are key developmental transitions that have been previously described and that have relevance for human disease.

The Discussion could be more informative. The manuscript does not include a lot of discussion – for example, you do not compare your strategy with what others have done and shown the novelty and the advantage of your method. You also do not consider the limitations of your approach? What were its weaknesses?

We thank the reviewer for raising this issue. We have revised the discussion to include comparisons, strengths, and weaknesses of our approach.

We already know a bit about the expression of the different basement membrane component and the switch – can you emphasise what is novel in your work? The expression of the components is not very novel – it is now available on line in KIT? What do your results add to this?

We appreciate this comment on the basis that aspects of the temporal and spatial expression of some basement membrane components during kidney development have been characterized in vivo (reviewed in PMID: 10594777, PMID: 23774818). The advance in our work is the global context with proteomics, which allows the comparative analysis of multiple basement membrane and other cellular components. This in turn allows the generation of hypotheses to explore the underlying mechanisms of developmental transitions and basement membrane regulation. We are also very familiar with the Kidney Interactive Transcriptomics (KIT) and these datasets have been extremely valuable to our developmental biology and kidney communities. In this manuscript we provide a comparative analysis across of single cell transcriptomic data from kidney organoids and in vivo mouse and human kidney samples. This is not possible by searching online interactive resources, and it required reanalysis of raw data. Therefore, our single cell data analysis will be of value to other researchers with a matrix and basement membrane focus

You probably do not need to abbreviate basement membrane to BM. Usually we now refer to collagen IV α3 chain rather than collagen α3(IV) chain.

To make the manuscript text accessible we use very few abbreviations but to avoid repeating the full term basement membrane exhaustively and sometimes within the same sentence, we have retained BM as an abbreviation. Although collagen a3(IV) is accepted terminology, we agree this is not as easy to follow and so have now changed to collagen IV αX chains.

A diagram of the different stages of kidney development would be useful…with the proteins in control and expressed in each.

Thank you for this suggestion. We have added a summary Figure 7 to illustrate the developmental milestones in kidney morphogenesis across the organoid, in vivo mouse, and human systems and related these to the sequence we observe in the assembly of core basement membrane components.

Usually we do not refer to people with a disease as a 'case'. This description of the boy could be much shorter. You have not indicated that you had ethics approval for this?

We have revised this terminology using tracked changes.

You need to describe pathogenic variants using the ACMG criteria to indicate that they are pathogenic. If P1096S is pathogenic then that complicates matters…

We have added the ACMG criteria as follows:

*1. COL4A5*: c.3695G>A; p.(Gly1232Asp): PM1_strong, PM2_moderate, PM5_moderate, PP3_moderate. Likely pathogenic.

*2. COL4A4*: c.3286C>T; p.(Pro1096Ser): PM2_moderate, PP3_moderate. Variant of uncertain significance.

Reviewer #2 (Recommendations for the authors):My key suggestion would be to bring out an emphasis that this is a model that best correlates and is applicable to kidney disease, as kidney BMs will have variation from other BMs. This is not a detraction from this work, I think this creates a new stimulus to investigate BMs across tissues and to create tractable models similar to the kidney organoids presented here.

As above we have made revisions to the text to highlight these points.

Reviewer #3 (Recommendations for the authors):The current submissions addresses temporal and tissue-specific BM changes during organoid kidney development. Day 25 kidney organoids contained tubules, stroma, and glomeruli with partial resemblance to (mouse) E19 kidney. Tubular and glomerular BMs are seen to form, the latter showing the expected switch from α-1/β-1 laminins to α-5/β-2 laminins, and alpha1/2 type IV collagens to alpha3-containing type IV collagens required for glomerular maturation.Specific questions/comments:1. Figure 2 and text: [a] Figure 2A, Microvascular endothelium: A key feature of the study in the replacement of Lm111 by Lm521 and Col-IV alpha1/2 by Col-IV α 3/4/5 in the developing glomerulus. This occurs in normal maturation as the glomerulus gains a vascular tuft. In this study, a regent specific to CD31 (PECAM-1; found on endothelial cells, but also on macrophages, granulocytes) is used to detect endothelial cells. Figure 2A shows increasing CD31 staining at days 11, 18 and 25. While most of the staining at day 25 appears between or at the edges of tubular-like structures, a single narrow stained-streak that co-localizes with laminin is present within a glomerulus. This may reflect the early appearance of a vascular tuft with a laminin coat. Are similar entities present in other glomeruli? If so, it would help to show this. Further, this reviewer feels it is important to show the ultrastructure of such glomeruli to gain insight into the organization (e.g. adjacent cells forming lumens) and surrounding structures of the C31-positive cells. Is there an early electron-dense BM structure lining these cells and, in particular, Are there both podocyte-derived and endothelial cell derived BMs?

Thank you for these comments. Overall, we conducted several immunofluorescence experiments, and the sequence and localization of basement membrane components was consistent. We also conducted additional electron microscopy of Day 25 organoids and we have included the images in a revised Figure 1 (panel 1D), and a dedicated Figure 1—figure supplement 2. These new images show both glomerular and tubular structures within the organoids and highlight basement membrane structures adjacent to podocyte primary processes. Regarding maturation of glomerular basement membrane structures in kidney organoids, we consider that this is limited by the absence of capillary blood flow. Indeed, we and others have shown further GBM maturation, including separate endothelial and podocyte basement membranes following organoid implantation into immunodeficient mice (PMID: 29429961, PMID: 2950308).

2. If vascular-tuft invasion of day 25 organoid glomeruli is only just beginning to occur (as suggested by the images), does it not follow that this organoid stage corresponds to an earlier mouse stage than that of the E19 mouse fetus? Given mice are born around E19.5 fully capable of forming urine, it is hard to imagine the implied correspondence. Perhaps the day 25 organoids reflect an earlier stage corresponding to the start of vascular invasion of the glomerulus (about E16.5).

We agree that overall, the day 25 organoid stage is closer to the earliest appearance of vascular invasion of the glomerulus and therefore E16 onwards. However, at E19 glomeruli are at different stages of development since the maturation process is not synchronized across all nephrons. This was our rationale for selecting the E19 time point in mouse development, since we expected to see a broader range of developing glomeruli. We then applied spatial proteomics using laser-capture to select consistent stages of glomerular development.

3. Figure 1G: An Alport patient iPSC line with an X-linked missense variant of COL4A5 was also evaluated and it was found that LAMB2 expression was increased in extra-glomerular BM. Such laminin compensation can apparently be seen in patients. The laminin beta2 increase appears to occur in all BMs of the organoids, not just the glomeruli. Is this what is seen in human patients?

An important question but very difficult to answer due to the paucity of relevant human tissue samples. Diagnostic genetic testing in Alport syndrome has superseded kidney biopsy in many Centres and although we have examined kidney biopsies from paediatric patients with Alport syndrome (PMID: 34049963) we have not examined laminin beta2 levels.

4. Figure 4. Distribution of BM components: [a] By day 18, laminin beta2 is seen to be present in the BMs of glomeruli and tubules. However, the expectation is that beta2 should largely be confined to glomeruli and blood vessels. BMs do not assemble on all cell surfaces as a generality (e.g. compare the basal side of epithelia with stromal fibroblasts). Further, specific BM components are only incorporated into select BMs. Such specificity can be a consequence of a lack of cognate receptors (especially relevant for laminins), other binding sites, and/or a consequence of a barrier to diffusion of secreted BM components from the source cells. So why is there a difference between mouse kidneys and organoid kidneys?

We agree that BM assembly is diverse and there is tissue specificity, that most likely links to BM function and to the BM receptors. The single cell transcriptomic data we present in Figure 4—figure supplement 1 highlights the variability in gene expression across cell types in kidney organoids. LAMB2 is expressed in several cell types but enriched in podocytes. The same is observed in the mouse and human kidney single cell sequencing analysis. Our immunofluorescence data also support enrichment over exclusivity of laminin beta2. A future analysis of the BM receptors in parallel with the BM ligands could help to further understand the predominant cell types responsible for BM component secretion.

[b] Laminin α 5 (Lma5) also appears to be widely distributed within the kidney organoid structures. During kidney development, Lma5 is strongly capillary loop/developing GBM localized (and also in blood vessels), although on can see some around tubules as well. In the organoids, the distribution appears quite even for glomerular, tubular and other structures.

As with the above response for laminin-β 2, the single cell transcriptomic data and immunofluorescence support enrichment versus exclusivity. Since we have characterized these systems with approaches and time points that have not been used before, it is possible that we are observing the wider variability in localization. For collagen IV α3 we saw enrichment in the Bowman’s capsule BM’s in fetal human kidney, whereas this is not described in mature human glomeruli.

One therefore has to wonder whether the same receptors (integrins, dystroglycan, other) are expressed in normal developing kidney as compared to organoid kidneys.

*[c]* Thank you for this suggestion. We have added new immunofluorescence data to demonstrate the colocalization integrin β 1 and laminin (Figure 1—figure supplement panel 1C). We also acquired and tested a dystroglycan antibody (DAG-6F4 from Developmental Studies Hybridoma Bank, DHSB) but were not able to detect an immunofluorescence signal with two experimental repeats. However, we detected dystroglycan in the organoid proteomics data all time points, increasing from day 14 to day 25 indicating the presence of this laminin receptor. The localization studies will require further optimization, but we do not think this data is essential for the interpretation of our manuscript.

[d] Collagen-IV alpha3 appears to be as widely distributed as collagen IV alpha1. Normally, Col4a345 is strongly localized to the capillary loops and GBM. The mechanism for the more restricted assembly of these subunits is not well understood as far as I am aware. Discussion of this would be helpful.

Extraglomerular localization of the collagen IV α3,4,5 network is reported in both mouse and human kidneys. However, the dominant signal is in the GBM. Similarly, we have observed the dominant signal in glomerular cell clusters. This distinction may well be more evident with maturity as we previously found clear glomerular specificity for the collagen IV α3,4,5 network compared to the collagen IV α1,1,2 network in organoids implanted into immunodeficient mice (PMID: 29429961).

5. Lines 70-72 of text. The authors state that "there is limited understanding about BM assembly". This reviewer would argue that actually there is a substantial body of published evidence derived from biochemical, cell biological and genetic approaches. This body is relevant to interpret the organoid/proteomic findings. In any case, different approaches address different aspects of BM assembly, extending from elucidation of binding interactions among components and with receptors to site-specific turnover. Renal organoid differentiation can reveal a host of new information; however, it probably is not the best tool to establish which components bind to each other via specific domain interactions, or the molecular mechanisms that underlie the interactions. To this reviewer, the real kidney advance of the study is a new way to understand the dynamic changes that occur during assembly in a temporal and cell-type specific manner.

We completely agree that there is a wealth of pioneering biochemical, cell biological and genetic studies that have laid the foundation for the field of basement membrane research. As such we have revised our statement to reflect the limited understanding in relation to kidney basement membranes. We also agree that the kidney organoid system has potential to understand the dynamics of BM assembly by the use of proteomics or reporter molecules.

6. It would be useful to enumerate both what the authors consider the strengths and limitations of the organoid kidney/proteomic system. It would also be useful to more explicitly state the advances achieved since the 2018 publication.

Thank you for this suggestion. We have revised the discussion to highlight the novel findings from this study.